# Two human brain systems micro-structurally associated with obesity

Manfred G Kitzbichler[1], Daniel Martins[2], Richard AI Bethlehem[1], Richard Dear[1], Rafael Romero-Garcia[1,3], Varun Warrier[1,4], Jakob Seidlitz[5,6,7], Ottavia Dipasquale[2], Federico Turkheimer[2], Mara Cercignani[8], Edward T Bullmore[1†], Neil A Harrison[8*†]

[1]Department of Psychiatry, University of Cambridge, Cambridge, United Kingdom; [2]Institute of Psychiatry, Psychology and Neuroscience, King's College London, London, United Kingdom; [3]Department of Medical Physiology and Biophysics, Instituto deBiomedicina de Sevilla (IBiS) HUVR/CSIC Universidad de Sevilla/ CIBERSAM, ISCIII, Sevilla, Spain; [4]Department of Psychology, University of Cambridge, Cambridge, United States; [5]Lifespan Brain Institute, The Children's Hospital of Philadelphia and Penn Medicine, Philadelphia, United States; [6]Department of Child and Adolescent Psychiatry and Behavioral Science,The Children's Hospital of Philadelphia, Philadelphia, United States; [7]Department of Psychiatry, University of Pennsylvania, Philadelphia, United States; [8]Brain Research Imaging Centre, Cardiff University, Cardiff, United Kingdom

*For correspondence:
harrisonn4@cardiff.ac.uk

†These authors contributed equally to this work

**Competing interest:** The authors declare that no competing interests exist.

**Abstract** The relationship between obesity and human brain structure is incompletely understood. Using diffusion-weighted MRI from ~30,000 UK Biobank participants, we test the hypothesis that obesity (waist-to-hip ratio, WHR) is associated with regional differences in two micro-structural MRI metrics: isotropic volume fraction (ISOVF), an index of free water, and intra-cellular volume fraction (ICVF), an index of neurite density. We observed significant associations with obesity in two coupled but distinct brain systems: a prefrontal/temporal/striatal system associated with ISOVF and a medial temporal/occipital/striatal system associated with ICVF. The ISOVF~WHR system colocated with expression of genes enriched for innate immune functions, decreased glial density, and high mu opioid (MOR) and other neurotransmitter receptor density. Conversely, the ICVF~WHR system co-located with expression of genes enriched for G-protein coupled receptors and decreased density of MOR and other receptors. To test whether these distinct brain phenotypes might differ in terms of their underlying shared genetics or relationship to maps of the inflammatory marker C-reactive Protein (CRP), we estimated the genetic correlations between WHR and ISOVF ($r_g$ = 0.026, $P$ = 0.36) and ICVF ($r_g$ = 0.112, $P < 9 \times 10^{-4}$) as well as comparing correlations between WHR maps and equivalent CRP maps for ISOVF and ICVF ($P < 0.05$). These correlational results are consistent with a two-way mechanistic model whereby genetically determined differences in neurite density in the medial temporal system may contribute to obesity, whereas water content in the prefrontal system could reflect a consequence of obesity mediated by innate immune system activation.

## Editor's evaluation

Kitzbichler et al. conducted a valuable large-scale study using the UK Biobank data to explore the relationship between brain tissue microstructure and obesity and provided convincing evidence for two coupled yet distinct brain systems mediating relationships between free water and neurite density as markers of inflammation with the genes enriched for innate immunity and specific neurotransmitter receptors. Major strengths include the innovative and expansive approach to understanding the genetic factors, neurotransmitters and potential mechanisms underlying observed

alterations in cortical thickness and gray matter volume in obesity. The scope of the work goes beyond most standard neuroimaging analyses and reveals coherent patterns linking neurite density and free water to relevant neuroinflammatory and neurotransmitter pathways.

## Introduction

Obesity has long been recognised as a preventable risk factor for cardiovascular and metabolic disorders such as heart disease and type-2 diabetes. More recently, it has also emerged as an important risk factor for neurodegenerative disorders, linked to both an increased risk of dementia and accelerated age-associated cognitive decline (*Sellbom and Gunstad, 2012*). Defined as the excessive accumulation of adipose tissue in the body (*González-Muniesa et al., 2017*), the worldwide prevalence of obesity has more than doubled in the last thirty years, making it one of the most important global public health challenges (*Yatsuya et al., 2014*).

To date, cross-sectional and longitudinal studies investigating effects of obesity on the brain have focused almost exclusively on macroscopic aspects of brain structure such as total grey matter volume and cortical thickness. Results in this field were often contradictory: although studies tended to report lower gray matter volume in relation to obesity, some have also observed null or positive associations as described in a meta-analysis by *García-García et al., 2019*, who noted that the likely reasons for this were heterogeneities in brain and obesity metrics, a wide variation in sample size, and poor statistical methodology.

However, the emerging consensus indicates that typically studies are reporting negative associations between obesity (particularly visceral obesity indexed by waist to hip ratio: WHR) and (smaller) total grey matter volume (*Cox et al., 2019*) and (thinner) cortical thickness (*Caunca et al., 2019*). Notably, this negative association between body mass index (BMI) and global grey matter volume has been substantiated in a recent large-scale study conducted in the UK Biobank involving 9652 participants (*Hamer and Batty, 2019*). Recent meta-analysis of voxel-based morphometry studies, including data from 5882 participants and a mega-analysis of 6,420 participants from the ENIGMA MDD working group, have also identified a consistent association of obesity with reductions in grey matter volume and cortical thickness in the medial prefrontal and orbitofrontal cortex and the temporal pole (*García-García et al., 2022*; *Opel et al., 2021*).

These associations between obesity and macroscopic features of grey matter structure have also been supported by longitudinal studies. For example, Franz et al. showed that by the age of 64 years, participants whose BMI steadily increased over forty years had thinner cortex in several frontal and temporal brain regions compared to those whose BMI was stable (*Franz et al., 2019*). Other longitudinal studies have shown associations between age-associated increases in BMI and grey matter reductions in the medial temporal lobe (entorhinal cortex and hippocampus) and cingulate cortex (*Arnoldussen et al., 2019*; *Bobb et al., 2014*). Together with the finding (*Opel et al., 2021*) of a significant age-by-obesity interaction on cortical thickness driven by lower thickness in older participants, this suggests that the negative impact of obesity on the brain accumulates over time.

Together, these studies provide robust evidence for an association between obesity and macrostructural features of brain anatomy such as grey matter volume and cortical thickness. However, changes in grey matter volume and cortical thickness can be driven by multiple different underlying processes and our understanding of the microstructural features that underpin this relationship remain largely unknown (*Westwater et al., 2022*). For example, it is currently not known whether obesity-associated differences in grey matter volume relate to changes in the size, shape or number of neurons e.g. neurite density or orientation dispersion within that region or alternately to differences in tissue water content. To date, the only studies to have investigated associations of obesity with brain microstructure have focused on white matter. Interestingly, these have identified obesity-associated differences in a number of different microstructural features of white matter including (1) obesity-related increases in white matter water content, (2) reduced myelination, and (3) lower fractional anisotropy (*Zhang et al., 2018*; *Kullmann et al., 2016*). However, whether comparable differences in cortical and subcortical grey matter micro-structure can be observed with obesity are yet to be reported.

We hypothesized that obesity would be associated with diffusion-MRI measures of grey matter tissue microstructure at 180 cortical regions and 8 subcortical structures (bilaterally) produced using neurite orientation dispersion and density imaging (NODDI) modelling of data from ~30,000

**eLife digest** People with obesity are at greater risk of cardiovascular diseases and metabolic conditions such as type 2 diabetes. More recently obesity has also been linked to changes in the brain that are associated with age-related dementia and cognitive decline. This includes a thinner cortex (the brain's outer layer) and lower volume of grey matter which is where cognitive processes, such as learning, take place.

However, questions remain about how obesity and grey matter are connected. For instance, it is unclear whether the change in volume is due to there being fewer cells (and thus more water between them) or fewer connections between cells in these brain areas. It is also unknown whether the reduced volume of grey matter is a cause or consequence of obesity.

To address these questions, Kitzbichler et al. analysed 30,000 MRI scans of the human brain which are stored in the UK Biobank. This revealed two characteristics in grey matter that were linked to obesity: higher amounts of water between cells in some areas, and a lower density of connections between neurons in others.

The areas with higher levels of free water are known to have more glial cells which provide support to neurons. They also have more receptors that bind to fatty acids (which are often raised in people with obesity) and more receptors for molecules and cells involved in the immune response. In contrast, the areas with a lower density of connections between neurons usually were more closely associated with genetic risk factors associated with obesity, and fewer receptors involved in feeding, appetite and energy use.

The findings of Kitzblicher et al. suggest that differences in the density of connections between neurons may contribute to obesity. High water content in grey matter, on the other hand, may be a consequence of obesity that occurs as a result of immune receptors becoming activated. This provides new insights in to how obesity and grey matter in the brain are connected.

participants in the UK Biobank MRI cohort. Unlike conventional diffusion MRI which models data acquired at a single diffusion weighting (shell), NODDI requires data collected at multiple different diffusion weightings (shells) then exploits the diffusion characteristics that can be observed in different tissue compartments to quantify their respective volume fractions. In this model, diffusion is modelled as isotropic in free water, restricted within neurites, and hindered in the extracellular space resulting in three microstructural metrics: Intracellular Volume Fraction (ICVF), which captures the volume fraction occupied by neurites (axons and dendrites) but not cell bodies; Orientation Dispersion Index (OD), which captures the spatial distribution of these processes; and isotropic volume fraction (ISOVF), which provides a measure of free water index.

Given previous findings of significant association between macroscopic differences in brain structure and visceral obesity, we elected to report associations with WHR in the main text and report complementary results for BMI as a measure of whole body obesity in the SI. Specifically, we tested each metric at each region for association with waist-to-hip ratio (WHR), and identified two anatomically and functionally distinct brain systems associated with obesity, using prior maps of gene expression, cellular composition and neurotransmitter receptor density to refine functional characterization of each obesity-associated system.

Finally, we then completed two further analyses to explore the potential directionality of the relationship between obesity and brain microstructure. In the first, we we used genome-wide association statistics (GWAS) for brain ISOVF and ICVF (*Warrier et al., 2022*), and for WHR (*Pulit et al., 2019*), to estimate the genetic correlations between each MRI metric and WHR, and test the secondary hypothesis that the WHR would have a tighter genetic correlation with ICVF than ISOVF. In the second, we produced brain maps for the association of ISOVF and ICVF with C-reactive protein (CRP), a measure of systemic inflammation. Given the pro-inflammatory properties of adipose (particularly visceral adipose) tissue we predicted tighter correlations between maps of CRP and ISOVF than maps of CRP with ICVF.

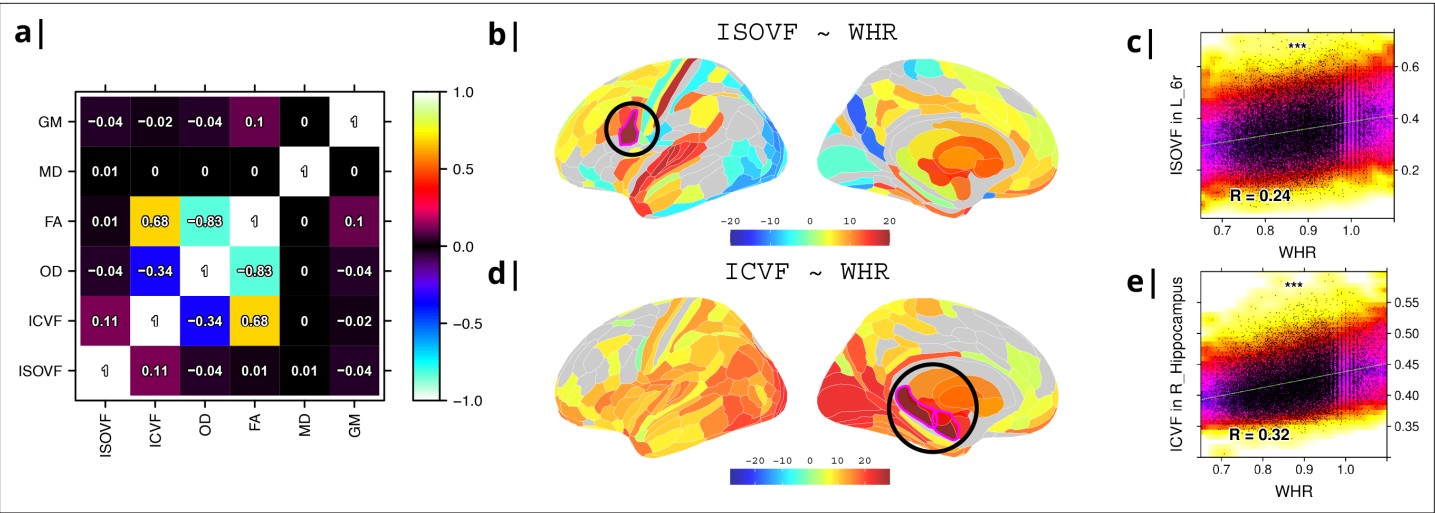

**Figure 1.** Micro-structural MRI metrics are associated with waist-to-hip ratio (WHR). (**a**) Correlation matrix for six macro- and micro/structural MRI metrics demonstrating that ISOVF (free-water) is essentially orthogonal to ICVF (neurite density) and OD, which instead form a cluster with FA. (**b**) Cortical and subcortical t-score map (left lateral and medial hemispheres) of ISOVF~WHR, representing the association of regional ISOVF with WHR, thresholded for significance at FDR = 5%. Circles indicate regions for which scatterplots are shown on the right. (**c**) Scatterplot of ISOVF in left inferior premotor region 6 r (y-axis) versus WHR (x-axis). (**d**) Cortical and subcortical t-score map of ICVF~WHR, thresholded at FDR = 5%. (**e**) Scatterplot of ICVF in the right hippocampus versus WHR. The maps of ISOVF~WHR and ICVF~WHR were negatively correlated ($r = -0.366$, $P = 2.3 \times 10^{-13}$). Colors in (**b and d**) refer to t-scores, colors in (c and e) denote normalised density. GM = Grey Matter; MD = Mean Diffusivity; FA = Fractional Anisotropy; OD = Orientation Dispersion Index; ISOVF = isotropic volume fraction; ICVF = intra-cellular volume fraction.

## Results

### Sample data

We used data provided by the UK Biobank, a population-based cohort of >500,000 subjects aged between 39 and 73 years (*Sudlow et al., 2015*) and focused on a subset of participants for whom complete multi-modal MRI data were available. Excluding participants with incomplete MRI data resulted in N ~30,000 participants for each dataset. For further details on participant numbers see *Appendix 2—table 2*.

### Association of waist-to-hip ratio with multimodal MRI measures of brain structure

Six MRI metrics of brain structure were used for correlational analysis with two measures of obesity (WHR and BMI) in N~30,000 participants from UK Biobank. WHR and BMI were strongly positively correlated with each other ($r = 0.428 \pm 0.009$, $P < 2 \times 10^{-16}$) and we therefore focus here on WHR although similar results are reported for BMI in Supplemental Information (see comparisons in *Appendix 2—figures 2 and 3* and *Appendix 2—figures 4 and 5* as well as *Appendix 2—figure 9*). Of the MRI metrics, there was one macro-structural measure (GM, grey matter volume) and five micro-structural measures (MD, mean diffusivity; FA, fractional anisotropy; OD, orientation dispersion; ICVF, intra-cellular volume fraction; and ISOVF, isotropic volume fraction). As illustrated in *Figure 1a*, some of these metrics were strongly correlated, indicating that they represented similar aspects of the underlying cortical micro-structure or tissue composition. For example, FA, OD, and ICVF metrics of neurite density were more strongly correlated with each other than with ISOVF, which is typically interpreted as a marker of tissue free water rather than cytoarchitectonics (*Kamiya et al., 2020*).

To address this potential redundancy, we performed a preliminary correlational analysis of all 6 MRI metrics with WHR then focused our subsequent analyses on ICVF and ISOVF, the two complementary MRI metrics that were most strongly associated with WHR. Comparable results for the other 4 metrics are reported in the Supplemental Information *Appendix 2—figure 2*.

Tissue free water (ISOVF) was significantly positively correlated with WHR (FDR = 5%) in 136 bilateral regions, concentrated in a prefrontal-temporal-striatal system comprising the prefrontal cortex (37 regions), superior temporal (primary auditory) cortex (21 regions), basal ganglia (caudate, putamen,

pallidum, accumbens), hypothalamus and thalamus. Referencing a database of prior task-related fMRI studies, this anatomical pattern of fMRI activations has been activated by tasks involving reward, auditory and musical functions (see *Appendix 2—figure 3b, c*). There were also some areas of significant negative correlation between ISOVF and WHR in the lateral and medial secondary visual cortex (see *Figure 1b*).

In contrast, neurite density (ICVF) was significantly positively correlated with WHR (FDR = 5%) in 152 bilateral regions concentrated in a medial temporal-occipital-striatal system comprising medial and lateral occipital cortex (26 regions), medial temporal lobe (hippocampus and amygdala), basal ganglia (putamen, pallidum, accumbens), hypothalamus and thalamus (see *Figure 1d*). This anatomical pattern has previously been activated by fMRI tasks involving episodic memory and navigation (see *Appendix 2—figure 3d, e*).

Maps of ISOVF~WHR and ICVF~WHR were negatively correlated ($r = -0.366, P = 2.3 \times 10^{-13}$); see Supplemental Information for correlation matrix of all MRI~WHR maps. This suggests that obesity is associated with coupled but anatomically distinct changes in measures of brain water and neurite density.

## Enrichment analysis of genes transcriptionally co-located with brain maps of association between obesity and brain water content, ISOVF–WHR, and between obesity and neurite density, ICVF–WHR

To investigate the basis for these associations of WHR with tissue water content (measured by ISOVF) and neurite density (measured by ICVF), we used human brain gene expression data from the Allen Brain Atlas to identify the individual gene transcripts that were most strongly co-located with each map. To do this, we independently tested 13,561 gene transcripts for significant spatial correlation with each map, that is ISOVF~WHR or ICVF~WHR, controlling for multiple comparisons entailed by whole genome analysis with FDR = 5% (*Figure 2*). Similar results were obtained by sensitivity analyses of co-location of weighted whole genome expression with maps of the correlations between MRI metrics and BMI instead of WHR; see *Appendix 2—figure 7*.

The tissue water content map (ISOVF~WHR) was significantly positively co-located with 1,031 gene transcripts and significantly negatively co-located with 1140 transcripts (FDR = 5%; spin permutation corrected). Enrichment analysis of the genes weighted by their spatial co-location with ISOVF~WHR identified 15 biological processes that were significantly under-represented, and 1 class that was positively enriched, with FDR = 5% to control for 29,687 biological processes and 11,110 molecular functions tested for enrichment. The most under-represented process was 'response to interleukin-6' and the most enriched process was 'pattern recognition receptor activity', both processes linked to the innate immune system. Other under-represented processes involved 'protein localisation to the Golgi apparatus', 'mitochondrial metabolism', 'taste receptor activity', and 'tau protein kinase activity'.

In contrast, the neurite density map (ICVF~WHR) was significantly positively co-located with 1,242 gene transcripts and significantly negatively co-located with 1354 transcripts (FDR = 5%; spin permutation corrected). Enrichment analysis of the genes weighted by their spatial co-location with ICVF~WHR identified 20 biological processes that were significantly negatively enriched, and 6 classes that were positively enriched, with FDR = 5% to control for 29,687 biological processes and 11,110 molecular functions tested for enrichment. The most negatively enriched process was 'peptidyl-asparagine modification' and the most positively enriched process was 'taste receptor activity'. Other negatively enriched processes included 'protein kinase C-activating G-protein-coupled receptor (GPCR) signalling pathway', 'fatty acid derivative binding', and 'glutamate receptor activity'.

The whole genome weights of association (vectors of correlations per gene) with ISOVF~WHR and ICVF~WHR were negatively correlated ($r = -0.615, P < 2.2 \times 10^{-16}$). Thus, the gene transcripts spatially co-located with ISOVF~WHR and ICVF~WHR maps are coupled but biologically distinct. The prefrontal-temporal-striatal system where ISOVF was positively correlated with WHR was co-located with gene transcripts enriched for innate immune and metabolic processes, whereas the medial temporal-occipital-striatal system where ICVF was positively correlated with WHR was co-located with transcripts enriched fo 'G-protein coupled receptor signalling', 'fatty acid derivative binding', and 'glutamate receptor activity'.

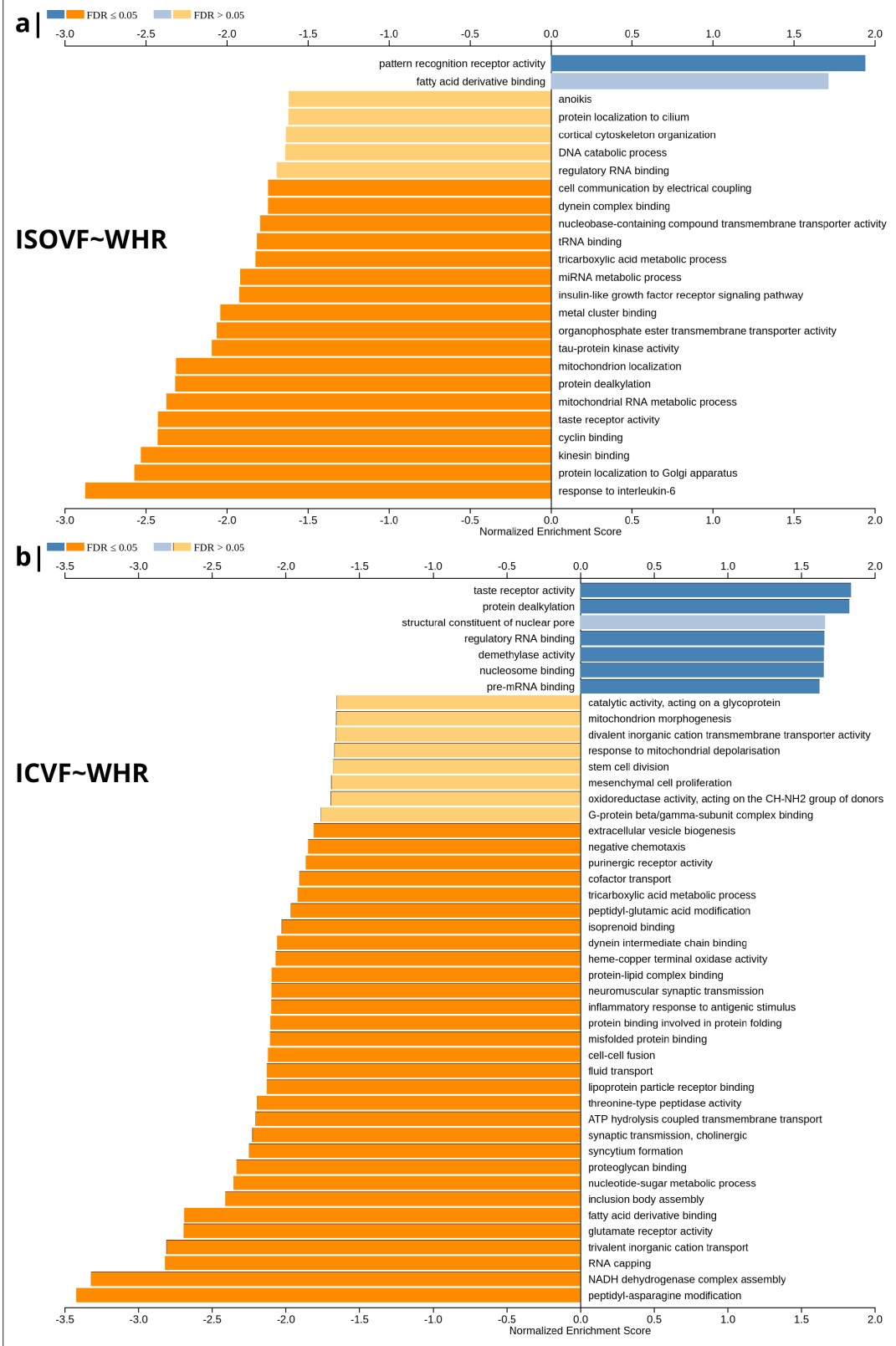

**Figure 2.** Significantly enriched gene ontology categories according to *Webgestalt* based on the spatial co-location of the MRI~WHR maps and whole brain expression maps for each of ~13,500 genes. (**a**) Results using the ISOVF~WHR maps (free water vs adiposity). Bar graph of significant gene ontologies showing normalized enrichment score on the x-axis. (**b**) Results using the ICVF~WHR maps (neurite density vs adiposity). In both cases,

*Figure 2 continued*

p-values for enrichment were tested by permutation taking into account the smoothness of cortical maps (using spin permutation correction; see *Appendix 2—figure 6*).

## Co-location of neurotransmitter and cellular atlases with brain maps of association between obesity and brain water content, ISOVF–WHR, and between obesity and neurite density, ICVF–WHR

To further investigate the brain systems where obesity was strongly associated with brain micro-structure measured by ISOVF or ICVF, we used prior data on human brain distribution of multiple neurotransmitter receptors *Hansen et al., 2022*. Cortical maps of each of 37 neurotransmitter receptors, for example, mu opioid receptor (MOR), were independently tested for spatial co-location with the ISOVF~WHR and ICVF~WHR maps, controlling for multiple comparisons with FDR = 5%.

The prefrontal-temporal-striatal system (ISOVF~WHR) was significantly (positively) co-located with the atlas distribution of five neurotransmitter receptors/transporters: 5HTT, serotonin transporter; D1, dopamine receptor; H3, histamine receptor; Mu, opioid receptor; and VAChT, acetyl-choline transporter.

In contrast, the medial temporal-occipital-striatal system (ICVF~WHR) was significantly (negatively) co-located with four neurotransmitter receptors/transporters: H3 histamine receptor; Mu opioid receptor; CB1 cannabinoid receptor; and A4B2, α4, β2 nicotinic acetyl-choline receptor.

Interestingly, the mu opioid receptor distribution was the most strongly correlated with both ISOVF~WHR and ICVF~WHR, but with opposite signs of association, meaning that regions where WHR correlated with neurite density typically expressed low mu opioid receptor density, whereas regions showing correlations between WHR and tissue water content typically expressed high mu opioid receptor density.

We likewise identified the cell-type distributions that were most strongly co-located with each of the ISOVF~WHR or ICVF~WHR maps. We independently tested 31 cell distributions atlases, provided by *Lake et al., 2018*, for significant spatial correlation with each map, controlling for multiple comparisons with FDR = 5%. The ISOVF~WHR map was significantly (positively) co-located with the atlas distribution of three glial cell classes: astrocytes, oligodendrocyte progenitor cells, and microglia. The ICVF~WHR map was not significantly co-located with any specific cell-type distribution.

## Genetic correlation analysis of obesity and micro-structural MRI phenotypes

The results reported above (and summarised in *Table 1*) indicate that obesity is associated with coupled changes in two anatomically, transcriptionally and neurobiologically differentiated brain systems, measured using ISOVF and ICVF micro-structural MRI metrics, respectively. On this basis we tested the hypothesis that genome-wide association statistics (GWAS) for normal

**Table 1.** Summary of differences between two obesity-associated micro-structural MRI phenotypes in terms of their associations with other brain phenotypes (gene ontology, receptor expression, and cell types) and their genetic correlations with obesity.

| | Scaling with obesity (WHR) | Gene ontology | Neurotransmitter receptors or transporters | Cell types | Genetic correlation with WHR |
|---|---|---|---|---|---|
| ISOVF (free water) | ↑ prefrontal-temporo-striatal system | ↑ pattern recognition receptors (PRR) ↑ receptors for fatty acid derivatives ↓ IL-6 responses | ↑ H3, Mu, D1 and 5HTT | ↑ astrocytes, microglia and oligodendrocyte precursor cells (not any class of neurons) | ○ not significant |
| ICVF (neurite density) | ↑ medial temporal-occipito-striatal system | ↑ taste receptor activity ↓ fatty acid receptors, glutamate receptor activity and GPCR signalling | ↓ H3, Mu, CB1 and A4B2 | ○ not significant | ↑ significant (positive) |

variation in ISOVF or ICVF (**Warrier et al., 2022**) were correlated with prior GWAS results for obesity (**Pulit et al., 2019**), indexed by WHR (see **Shungin et al., 2015**). We used linkage disequilibrium score (LDSC) analysis to estimate genetic correlations between WHR and ISOVF or ICVF. We found a modest, statistically significant positive genetic correlation between ICVF and WHR ($r_g = 0.11 \pm 0.030, P < 9 \times 10^{-4}$), but no genetic correlation between ISOVF and WHR ($r_g = -0.026 \pm 0.03, P = 0.3$); see **Appendix 2—table 2** for details. These results indicate shared effects of genetic variation on obesity (WHR) and neurite density (ICVF), but no shared genetic effects on obesity and brain water content (ISOVF).

### Relationship with peripheral inflammation

In the final analysis we compared the effects on microstructure of three variables of interest at the same time, WHR, BMI, and specifically CRP, a measure of systemic inflammation. To this end we looked at the pairwise relationships of the maps ISOVF~CRP vs ISOVF~WHR, ICVF~CRP vs ICVF~WHR, etc. (see **Appendix 2—figure 13**). Given the pro-inflammatory properties of adipose (particularly visceral adipose) tissue, for CRP we expected tighter correlations between ISOVF maps than ICVF maps. This hypothesis is indeed supported by our findings, the correlation is significantly stronger for the ISOVF maps than the ICVF maps (CRP-BMI: $P < 1.2 \times 10^{-5}$, CRP-WHR: $P < 0.024$, one-tailed). We also find that the WHR and BMI maps are different (WHR-BMI: $P < 0.05$, two-tailed).

## Discussion

Here, we have reported evidence, consistent with our first hypothesis, that obesity is associated with coupled changes in two micro-structural MRI metrics (ISOVF, free water; and ICVF, neurite density) in two anatomically, transcriptionally and neurobiologically differentiated brain systems. We have also reported genetic correlation analysis that was consistent with our secondary hypothesis, that these two distinct brain phenotypes have different genetic relationships with obesity.

### Obesity and brain MRI phenotypes

Previous well-powered studies have identified associations between obesity and a pattern of reduced grey matter volume or cortical thickness centred on fronto-temporal cortex and sub-cortical structures. Here, using NODDI modelling of diffusion-weighted MRI data from ~30,000 participants in the UK Biobank we have extended these findings to demonstrate associations between obesity (WHR) and two measures of grey matter microstucture, ISOVF (an index of tissue water content) and ICVF an index of neurite density (see **Table 1** for a summary).

Similar to previously reported associations with brain grey matter macrostructure, positive scaling of WHR and tissue water content (i.e. oedema) was most pronounced within frontal and temporal cortices and subcortical structures. In contrast, we observed a more anterior-posterior pattern of association between WHR and neurite density, with more obese individuals having higher neurite density in posterior compared to anterior brain regions. By relating obesity associated grey matter microstructure maps to gene expression data from the Allen Brain Atlas, we show that regions where WHR was more tightly linked to tissue water content had greater expression of pattern recognition receptors (PRR) and receptors for binding fatty acid derivatives, and reduced expression of genes associated with biological processes linked to interleukin-6 (IL-6) responses. Interestingly, these regions were also richer in astrocytes, microglia and oligodendrocyte precursor cells but not any class of neurons; and had high concentrations of some but not all neurotransmitter receptors or transporters tested, for example, histamine (H3), mu-opioid, D1, and 5HTT.

In contrast, the medial temporal-occipital-striatal system where obesity was associated with increased neurite density was co-located with expression of transcripts positively enriched for taste receptor activity and lower fatty acid binding, glutamate receptor activity and other biological processes linked to protein kinase C-activating G-protein-coupled receptor signalling. Interestingly, this system was not co-located with any specific cell class but it was co-located with specific neurotransmitter receptor maps including H3, Mu, CB1, and A4B2, meaning that regions showing the greatest positive scaling between neurite density and WHR showed relatively low expression of receptors linked to feeding, appetite, and energy expenditure.

## What are the potential causal relationships between obesity and brain MRI phenotypes?

ISOVF and ICVF are weakly correlated (i.e. independent) markers of free water and neurite density, respectively. Both are significantly and mostly positively correlated with WHR in brain systems. Obesity-related differences in ISOVF and ICVF were coupled (negatively correlated) but also anatomically, transcriptionally, and neurobiologically differentiated from each other (*Table 1*). This raises the question: Could these two brain phenotypes have a different causal relationship with obesity?

For example, it is conceivable that the changes in brain water associated with obesity could represent an effect of obesity on the brain, that is WHR→ISOVF, whereas the obesity-related changes in neurite density could represent an effect of the brain on obesity, that is ICVF→WHR. Such a bi-directional mechanistic model of the relationships between obesity and the brain seems somewhat plausible. Obesity is usually caused by changes in eating behaviour and physical activity, which are controlled by brain systems enriched for opioid, dopamine and cannabinoid receptor-mediated signalling. So changes in the brain, indexed by neurite density, could conceivably cause adipogenic eating behaviours and thus obesity. Obesity in turn causes a pro-inflammatory state systemically and blood concentrations of CRP, IL-6 and other cytokines have previously been associated with changed (increased) micro-structural MRI metrics of free water (*Kitzbichler et al., 2021*). So inflammation could potentially mediate effects of obesity on the brain tissue water content (see also *Turkheimer et al., 2022*). Our finding that the CRP-WHR map correlation is significantly stronger for the ISOVF maps than the ICVF maps would be consistent with this hypothesis.

Using novel techniques for analysis of spatial co-location of whole genome transcript maps and MRI phenotypes to optimise subsequent enrichment analysis of strongly co-located gene transcripts, we found that transcripts co-located with ISOVF~WHR were enriched for IL6 and pattern recognition receptors (PRRs), both implicated in innate immune signalling; whereas transcripts co-located with ICVF~WHR were enriched for taste receptors. This pattern of results is consistent with the model that changes in neurite density associated with obesity might reflect primary brain changes in taste sensation and reward processing that drive consummatory behaviours leading to obesity; whereas changes in brain free water associated with obesity might reflect effects of pro-inflammatory cytokines produced by adipose tissue that drive extravasation and oedema in some brain regions.

One limitation of this study is that data was collected at multiple centres and even though we used site as a nuisance regressor there might be unaccounted for non-linear effects. However *Duff et al., 2022* showed that quantities derived from UK Biobank scans at different sites are reliable.

It should also be mentioned that the age range of the AHBA donors (24-57 years) is only partially overlapping with the participants in the UK Biobank (44-80 years). Future studies will hopefully provide a more comprehensive picture of whole brain gene expression as a function of age so that the powerful strategy for linking transcriptional and imaging data that the AHBA dataset has enabled can be extended to gene expression datasets more closely aligned demographically with the neuroimaging dataset of interest. These and other methodological issues relating to alignment of AHBA gene expression data with MRI phenotypes have been rigorously reviewed in detail (*Fornito et al., 2019*; *Arnatkeviciute et al., 2023*).

Concerning the question whether both brain systems are in operation in the same individual at the same time, we are not aware of any currently available tools that would allow us to actually test this assumption, but it could be an interesting avenue for future work. Another limitation of our study is that it is based on a cross-sectional dataset, and it is therefore impossible to disentangle causally directed relationships with certainty from correlations between MRI and transcriptional phenotypes. We also approached this question by using GWAS data on obesity and each of the two MRI metrics to estimate and test genetic correlations between obesity and ISOVF or ICVF. We found that ICVF was genetically correlated with obesity, but not ISOVF. This result is consistent with the bidirectional mechanistic model, whereby changes in neurite density (but not brain water) cause obesity, but it does not prove it. There are many other possible interpretations of a genetic correlation between phenotypes, that is pleiotropic genetic effects on both phenotypes, which do not entail a causal relationship between phenotypes. Further work will be needed to validate this and other causal models of the directional relationships between obesity and the brain, which could be important for future prevention, diagnosis, and treatment of obesity.

# Materials and methods

## Data available in UK Biobank

### Participants

Data were provided by the UK Biobank (application IDs 20904 & 48943), a population-based cohort of >500,000 subjects aged between 39 and 73 years (*Sudlow et al., 2015*). We focused on a subset of N = 40,680 participants for each of whom complete multimodal MRI data were available for download (February 2020). We excluded participants with incomplete MRI data resulting in the numbers for each dataset shown in *Appendix 2—table 2*.

### Imaging data acquisiton

Minimally processed T1- and T2-FLAIR- weighted MRI data (and DWI data) were downloaded from UK Biobank (https://biobank.ctsu.ox.ac.uk/crystal/crystal/docs/brain_mri.pdf). The acquisition of these MRI data has been described in detail in *Alfaro-Almagro et al., 2018*, and is summarised here. MRI data at all three sites were collected on a 3T Siemens Skyra scanner (Siemens, Munich, Germany) using a 32-channel receive head coil. T1-weighted images were acquired using a 3D MPRAGE sequence with the following key parameters; voxel size 1mm × 1mm × 1mm, TI/TR = 880/2000 ms, field-of-view = 208 × 256 × 256 matrix, scanning duration = 5 min. The diffusion weighted imaging data were acquired using a monopolar Steejskal-Tanner pulse sequence and multi-shell acquisition (b=0 s/mm$^2$, b=1.000 s/mm$^2$, b=2.000 s/mm$^2$) with the following key parameters; voxel size 2mm × 2mm × 2mm, TE/TR = 92/3600 ms, field-of-view = 104 × 104 × 72 matrix, and scanning duration = 7 minutes (*Alfaro-Almagro et al., 2018*).

## Imaging pre-processing

### Structural MRI

Minimal processing for T1-weighted data included defacing, cutting down the field-of-view and gradient distortion correction using Brain Extraction Tool (*Smith, 2002*) and FLIRT (FMRIB's Linear Image Registration Tool) (*Jenkinson et al., 2002*). The data were then nonlinearly warped to MNI152 space using FNIRT (FMRIB's Nonlinear Image Registration Tool) (*Andersson and Sotiropoulos, 2016*). Next, tissue-type segmentation was done using FAST (FMRIB's Automated Segmentation Tool) (*Zhang et al., 2001*) and a bias-field-corrected version of the T1 was generated (*Alfaro-Almagro et al., 2018*).

### Further processing

We used these data as input to Freesurfer V6.0.1 (*Fischl et al., 2004*) using the T2-FLAIR weighted images to improve pial surface reconstruction. Following reconstruction, the Human Connectome Project (HCP) parcellation (*Glasser et al., 2016*) was aligned to each individual image and regional metrics were estimated for 180 bilateral cortical areas and eight bilateral subcortical structures (giving a total of 376 areas).

### Diffusion weighted MRI

Minimal processing for diffusion weighted imaging (DWI) data included correction for eddy currents (*Andersson and Sotiropoulos, 2015*; *Andersson and Sotiropoulos, 2016*), head motion, outlier-slices removal and gradient distortion correction (*Alfaro-Almagro et al., 2018*).

### Further processing

We then co-registered the DWI data with the T1-aligned parcellation template to estimate fractional anisotropy (FA) and mean diffusivity (MD) at each region using DTIFIT [https://fsl.fmrib.ox.ac.uk/fsl/fslwiki/FDT/UserGuide#DTIFIT]. For each scan, the first B0 image of the diffusion-sensitive sequence was linearly coregistered to the T1 image with FLIRT. The resulting inverse transformation was used to map the parcellation into the DWI space. Neurite orientation dispersion and density imaging (NODDI) reconstruction was done using the AMICO pipeline (*Daducci et al., 2015*). Documentation and code for these processing pipelines is available on Github (https://github.com/ucam-department-of-psychiatry/UKB, copy archived at *Romero-Garcia, 2023*).

## Imaging quality control

We used T1-weighted and T2-weighted scans for the Freesurfer anatomical image reconstruction, because this approach improves anatomical reconstruction (*Glasser et al., 2013*). However, subjects

without T2 scans had cortical thickness systematically biased towards lower values compared to subjects with both T1 and T2 images. Thus, we excluded participants without T2 scans from all analyses. In order to avoid spurious effects from pathologies causing systemic inflammation, we also excluded subjects with high CRP (>). We repeated the analysis without subjects who had reported an episode of stroke or diagnosis of dementia, producing identical results.

## Analysis pipeline

A detailed description of the full processing pipeline can be found in Supplemental Information Appendix 2; briefly, it comprised the following steps: Load and match UKB imaging data with socio-demographic and health data. Regress imaging modalities from NODDI dataset onto waist-to-hip ratio (WHR) with age, sex, scan quality, and scan site as nuisance regressors. This is done for males and females at the same time, but including sex as a covariate (for sensitivity analysis separating by sex see *Appendix 2—figure 3*). Adopting the pseudo-code format used by the *R* statistical language, the regression formula was: $\text{ISOVF} + \text{ICVF} \sim \text{WHR} + \text{Age} + \text{Sex} + \text{Quality} + \text{Site}$ where *Quality* is quantified by the Freesurfer Euler number (a higher number means more surface reconstruction errors) and *Site* was one of three sites encoded as categorical variable.

The terms on the left can be represented as matrices having $N_{subjects}$ rows and $N_{ROIs}$ columns, whereas the terms on the right are vectors with $N_{subjects}$ entries. Then for each term on the left (ie. imaging modality) the result is a matrix of t-statistics or p-values with dimension $N_{covariates} \times N_{ROIs}$. The relevant row from this matrix is the one relating to the WHR coefficient which can be plotted as a brain map as shown in *Figure 1* and *Appendix 2—figure 3* for each imaging modality, respectively.

### ABAGEN gene expression maps

We then related these maps to anatomically localized gene expression data from the Allen Brain Atlas (*Hawrylycz et al., 2012*) using the *ABAGEN* package (*Markello et al., 2021*) to map gene expression onto the same parcellation as the imaging data (Glasser HCP). The 43 (predominantly small) regions without gene expression data were excluded from analysis and are grayed out on the brain maps.

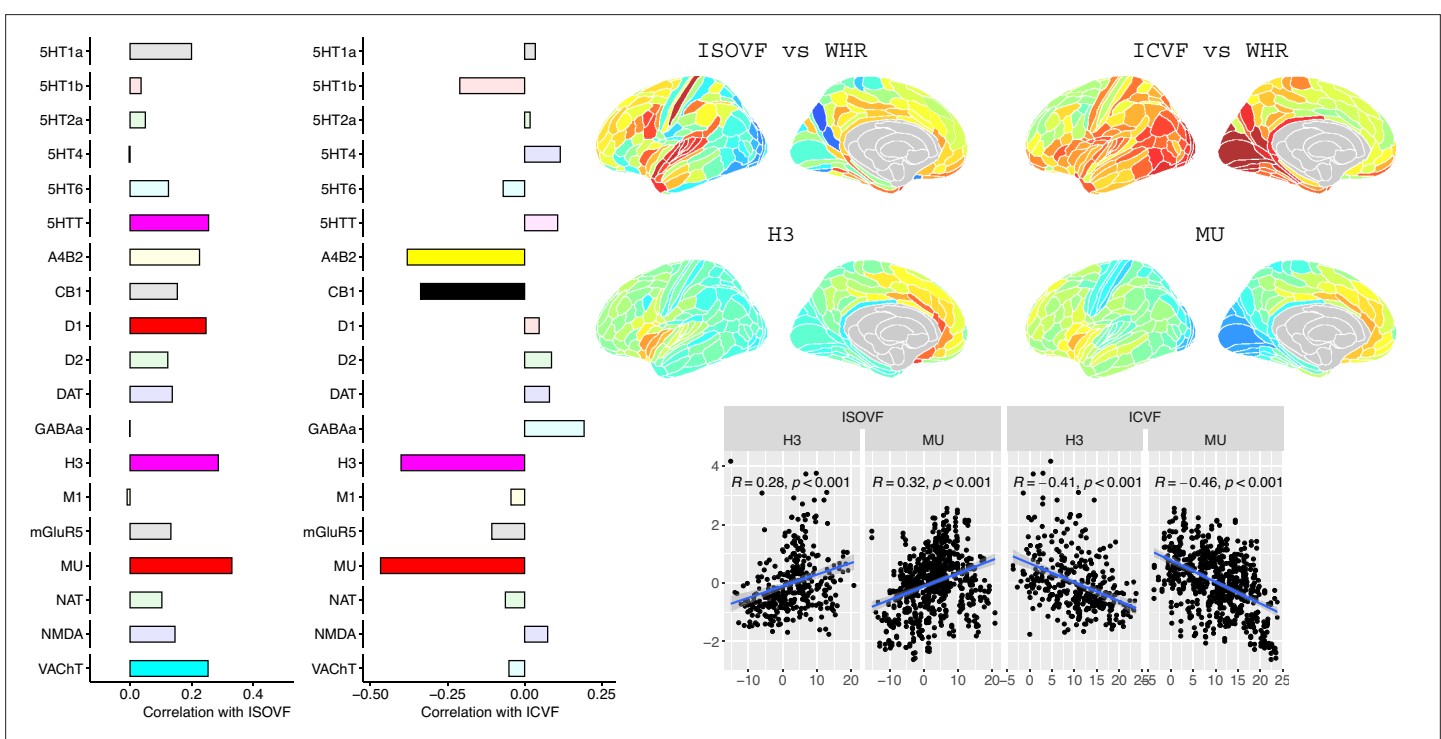

**Figure 3.** Co-location of neurotransmitter receptor or transporter distributions with obesity-associated micro-structural MRI systems. Left: Correlations of cortical neurotransmitter maps with the ISOVF~WHR and ICVF~WHR maps shown above (same color scale as in *Figure 1*). Significance is indicated by shading (based on spin permutation and Bonferroni correction). The Mu and H3 receptors show the maximum (absolute) correlation with the ISOVF and ICVF maps of microstructural effect of obesity (top right). Bottom right: scatter plots of raw data.

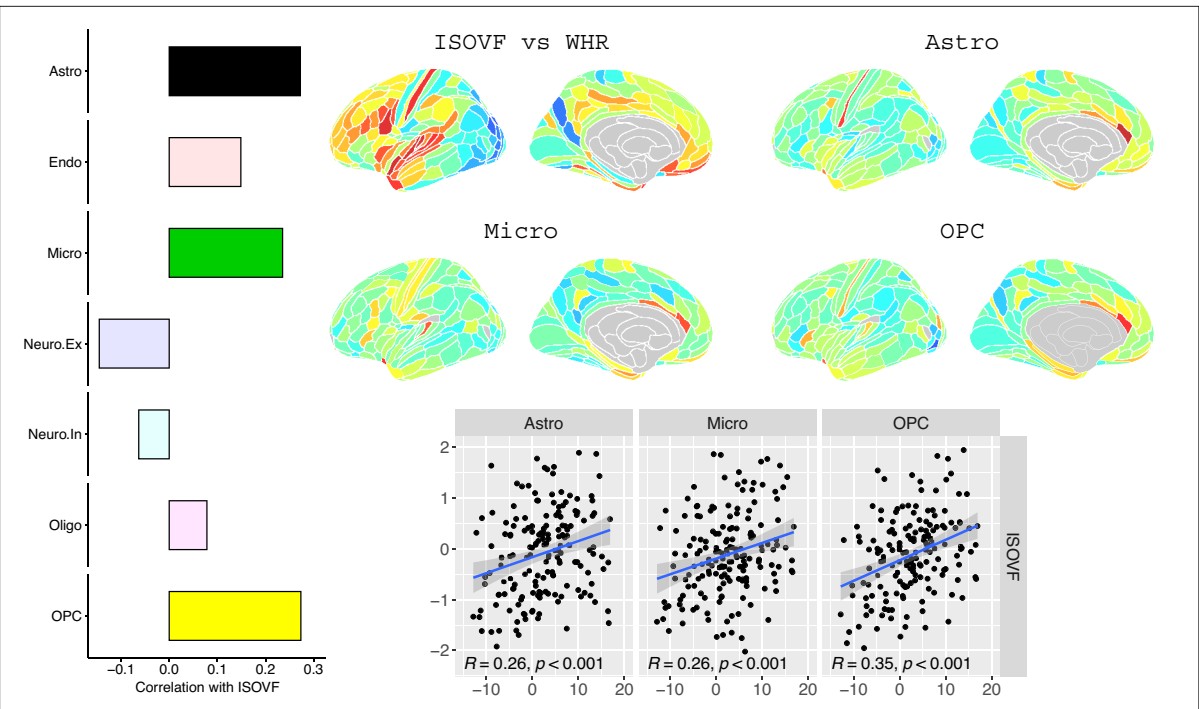

**Figure 4.** Co-location of brain cell distributions with obesity-associated micro-structural MRI systems. Left: Correlations of brain cell type maps for seven cell type categories from *Lake et al., 2018* with the ISOVF~WHR map shown above (same color scale as in *Figure 1*). Significance is indicated by shading (based on spin permutation and Bonferroni correction). The astrocytes, microglia, and OPC cell type maps show the maximum (absolute) correlation with the ISOVF~WHR maps (central panel). Right: scatter plots of raw data. (Results for ICVF were not significant for any category and are only shown in the Supplemental Information.).

Subsequently, we performed a correlation analysis with the ABAGEN maps (~13,500 maps, one for each gene) as predictors and the NODDI-WHR maps as responses. We then repeated this step 1000 times for spin-permuted versions of the NODDI-WHR maps to generate a set of 1000 surrogate gene correlations. This was done separately but in parallel for both ISOVF and ICVF. The resulting real and surrogate data loadings were input to the gene enrichment analysis tool *Webgestalt* (*Wang et al., 2017*), which was modified to incorporate the spin permutation process instead the default process of random permutations to calculate p-values. This yielded a number of significantly enriched gene ontology categories with FDR corrected $P_{FDR} < 0.05$, as shown in *Figure 2*. Supplemental Information *Appendix 2—figure 6* contains a schematic of the analysis pipeline.

## Neurotransmitter maps

*Hansen et al., 2022* compiled 37 neurotransmitter receptor atlases from the literature and provided them as 3D volumes in MNI space. These were then parcellated in the same way as the imaging data (using the Glasser HCP template). We independently tested the resulting 37 neurotransmitter maps (*Appendix 2—figure 11*) for significant spatial correlation with the MRI~WHR maps, controlling for spatial autocorrelation using 10,000 spin permutations and correcting for multiple comparisons with FDR = 5%. Maps for the same receptor from different literature sources were correlated independently but the results were combined, resulting in the 19 separate receptors shown in *Figure 3* (see *Appendix 2—figure 9* for a sensitivity analysis using the original 37 maps individually).

## Cell type maps

*Lake et al., 2018* provided 31 brain cell distributions atlases based on single-cell DNA transcription analysis. These were then parcellated in the same way as the imaging data (using the Glasser HCP template). We independently tested the resulting 31 cell-type maps (*Appendix 2—figure 12*) for significant spatial correlation with the MRI~WHR maps, controlling for spatial autocorrelation using 10,000 spin permutations and for multiple comparisons with FDR = 5%. We concentrated on the

seven categories at the highest level (Astro, Endo, Micro, Neuro.Ex, Neuro.In, Oligo, OPC; *Figure 4*) and did not separately analyse the individual excitatory and inhibitory neuronal sub-types (Ex1-8 and In1-8).

## Genetic correlation analysis

We used genome-wide association statistics for ICVF and ISOVF (*Warrier et al., 2022*), and for waist-to-hip ratio (plain and adjusted for BMI; *Pulit et al., 2019*). Genetic correlations were estimated using linkage disequilibrium (LD) score regression (*Bulik-Sullivan et al., 2015*) based on LD information from North-West European populations.

## Acknowledgements

We are very grateful to Linda Pointon for organisational support. This research has been conducted using the UK Biobank Resource under Application Numbers 20904 and 48943. This study was funded by an award from the Wellcome Trust (grant number: 104025/Z/14/Z) for the Neuroimmunology of Mood Disorders and Alzheimer's Disease (NIMA) consortium (MGK, FT, MC, ETB, NAH). For the purpose of open access, the author has applied a CC BY public copyright licence to any Author Accepted Manuscript version arising from this submission. Additional support was provided by the National Institute of Health Research (NIHR) Cambridge Biomedical Research Centre (ETB). ETB was also supported by an NIHR Senior Investigator award.

## Additional information

### Funding

| Funder | Grant reference number | Author |
| --- | --- | --- |
| Wellcome Trust | 104025/Z/14/Z | Manfred G Kitzbichler<br>Federico Turkheimer<br>Mara Cercignani<br>Neil A Harrison |

The funders had no role in study design, data collection and interpretation, or the decision to submit the work for publication. For the purpose of Open Access, the authors have applied a CC BY public copyright license to any Author Accepted Manuscript version arising from this submission.

### Author contributions

Manfred G Kitzbichler, Conceptualization, Resources, Data curation, Software, Formal analysis, Validation, Investigation, Visualization, Methodology, Writing – original draft, Writing – review and editing; Daniel Martins, Conceptualization, Methodology; Richard AI Bethlehem, Data curation, Software, Funding acquisition, Writing – review and editing; Richard Dear, Varun Warrier, Software, Formal analysis, Methodology, Writing – review and editing; Rafael Romero-Garcia, Jakob Seidlitz, Data curation, Software, Writing – review and editing; Ottavia Dipasquale, Conceptualization; Federico Turkheimer, Conceptualization, Writing – review and editing; Mara Cercignani, Conceptualization, Methodology, Writing – review and editing; Edward T Bullmore, Neil A Harrison, Conceptualization, Supervision, Funding acquisition, Methodology, Writing – original draft, Writing – review and editing

### Author ORCIDs

Manfred G Kitzbichler ⓘ http://orcid.org/0000-0002-4494-0753
Daniel Martins ⓘ http://orcid.org/0000-0002-0239-8206
Richard AI Bethlehem ⓘ http://orcid.org/0000-0002-0714-0685
Mara Cercignani ⓘ http://orcid.org/0000-0002-4550-2456
Neil A Harrison ⓘ https://orcid.org/0000-0002-9584-3769

### Decision letter and Author response

Decision letter https://doi.org/10.7554/eLife.85175.sa1
Author response https://doi.org/10.7554/eLife.85175.sa2

# Additional files

## Supplementary files
• MDAR checklist

## Data availability
Data were provided by the UK Biobank (https://www.ukbiobank.ac.uk/, application IDs 20904 and 48943). Source code can be found on GitHub under https://github.com/ucam-department-of-psychiatry/UKB (copy archived at *Romero-Garcia, 2023*).

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

# Appendix 1

## Imaging data acquisiton

MRI data was collected on a 3T Siemens Skyra scanner (Siemens, Munich, Germany) using a 32-channel receive head coil. T1-weighted images were acquired using a 3D MPRAGE sequence with the following key parameters; voxel size 1mm × 1mm × 1mm, TI/TR = 880/2000 ms, Field-of-view = 208 × 256 × 256 matrix, scanning duration: five minutes. The diffusion weighted imaging data was acquired using a monopolar Steejskal-Tanner pulse sequence and multi-shell acquisition (b=0 s/mm$^2$, b=1.000 s/mm$^2$, b=2.000 s/mm$^2$) with the following key parameters; voxel size 2mm × 2mm × 2mm, TE/TR = 92/3600 ms, Field-of-view = 104 × 104 × 72 matrix and scanning duration = seven minutes (*Alfaro-Almagro et al., 2018*).

## Imaging preprocessing

We obtained T1 and T2-FLAIR weighted data from the UK Biobank after structural minimal processing. Minimal processing for T1 weighted data included defacing, cutting down the field-of-View and gradient distortion correction using Brain Extraction Tool (*Smith, 2002*) and FLIRT (FMRIB's Linear Image Registration Tool; *Jenkinson et al., 2002*). The data was then nonlinearly warped to MNI152 space using FNIRT (FMRIB's Nonlinear Image Registration Tool; *Andersson et al., 2007*). Next, tissue-type segmentation is applied using FAST (FMRIB's Automated Segmentation Tool; *Zhang et al., 2001*) and a bias-field-corrected version of the T1 is generated (*Alfaro-Almagro et al., 2018*). Minimal processing for Diffusion MRI data included correction for eddy currents (*Andersson and Sotiropoulos, 2015*; *Andersson and Sotiropoulos, 2016*), head motion, outlier-slices removal and gradient distortion correction (*Alfaro-Almagro et al., 2018*).

## Imaging quality control

We used T1-weighted and T2-weighted scans for the freesurfer anatomical image reconstruction, because this approach improves anatomical reconstruction (*Glasser et al., 2013*). However, subjects without T2 scans had cortical thickness systematically biased towards lower values compared to subjects with both T1 and T2 images. Thus, we excluded participants without T2 scans from all analyses.

## Genetic correlation analysis

We conducted genetic correlations using genome-wide summary statistics for ICVF and ISOVF (*Warrier et al., 2022*) as well as waist-to-hip ratio (plain and adjusted for BMI; *Pulit et al., 2019*). Genetic correlations were conducted using LD score regression (*Bulik-Sullivan et al., 2015*) based on LD information from North-West European populations.

# Appendix 2

## Analysis pipeline

- load and match UKB imaging data with sociodemographic and health data
- regress imaging modalities from NODDI dataset onto WHR with age as nuisance regressor and dropping subjects with excessive CRP and no T2. This is done for males and females at the same time with sex as nuisance regressor:

$$\text{ISOVF} + \text{ICVF} \sim \text{WHR} + \text{Age} + \text{Sex} + \text{Euler} + \text{Site} \quad \ldots \quad \forall\, \text{CRP} \leq 10 \wedge \exists\, \text{T2 scan}$$

- the resulting statistics for the WHR coefficient can be plotted as a brain map separately for each imaging modality as shown in *Figure 1*.
- add Allen Brain Atlas gene expression data to the mix
- use *ABAGEN* package (*Markello et al., 2021*) to map gene expression onto same parcellation as previous imaging data (Glasser HCP):

    - after matching samples to regions, only keep regions that have at least one samplefrom at least one of the six donors (43 regions did not)

  The other parameters used are:

    - filter out subcortical samples upfront using AHBA annotations of samples
    - use *Arnatkeviciute et al., 2019* for native parcellation images mapped to each of the six donor brains
    - when multiple probes are available for a gene, use them probe with highest differential stability (=mean correlation over spatial regions between all pairs of donors)
    - average samples into regions first within each donor separately, then across donors
    - normalize all samples to have same mean expression over genes, then normalize genes to have same mean expression over samples, both using scaled robust sigmoid method (see *Arnatkeviciute et al., 2019*)

- do correlation analysis with the ABAGEN maps (~13,000 maps, one for each gene) on the right (predictors $X$) and the NODDI-WHR maps on the left (responses $Y$):

    - as a sensitivity analysis, the process was repeated for BMI instead of WHR (*Appendix 2— figure 4*), and correlation was substituted by PLS regression. Statistical significance was tested by performing 1000 spin permutations of the ABAGEN data ($X$) and 1000 boot-strap resamples of the imaging data ($Y$). The explained variance per component for both $X$ (*Appendix 2—figure 5e*) and $Y$ (*Appendix 2—figure 5f*) is significantly higher for the empirical dataset (red) compared to the surrogate data distribution (boxes).

- feeding the loadings from the correlation analysis into the gene enrichment analysis tool *Webgestalt* (*Wang et al., 2017*) yielded a number of significantly enriched gene ontology categories (at spin and FDR corrected $P_{\text{FDR}} < 0.05$) as shown in *Figure 2*. The analysis was done separately but in parallel for ISOVF and ICVF.

**Appendix 2—table 1.** UK Biobank data.

| Variable | N | Female | Male |
|---|---|---|---|
| Age | 34,229 | 18,143 | 16,086 |
| Body Mass Index (BMI), kg/m² | 33,090 | 17,501 | 15,589 |
| Waist to Hip Ratio (WHR) | 33,183 | 17,560 | 15,623 |
| Visceral Adipose Tissue (VAT) | 7539 | 3957 | 3582 |
| Extracellular free water (isotropic volume fraction ISOVF) | 34,194 | 18,126 | 16,068 |
| Intracellular neurite density (intracellular volume fraction ICVF) | 34,194 | 18,126 | 16,068 |
| Intracellular neurite dispersion (orientation dispersion OD) | 34,194 | 18,126 | 16,068 |
| Fractional anisotropy (FA) | 34,194 | 18,126 | 16,068 |

| Variable | N | Female | Male |
|---|---|---|---|
| Mean diffusivity (MD) | 34,194 | 18,126 | 16,068 |
| Gray matter volume (GM) | 34,229 | 18,143 | 16,086 |

**Appendix 2—table 2.** Gene correlation analysis results.

| *trait 1 | †trait 2 | $r_g$ | std error | z-score | p-value |
|---|---|---|---|---|---|
| WHR | ISOVF | 0.0259 | 0.0282 | 0.9184 | 0.3584 |
| WHR | ICVF | 0.1118 | 0.0337 | 3.3187 | $9 \times 10^{-4}$ *** |

*from **Pulit et al., 2019**.
†from **Warrier et al., 2022**.

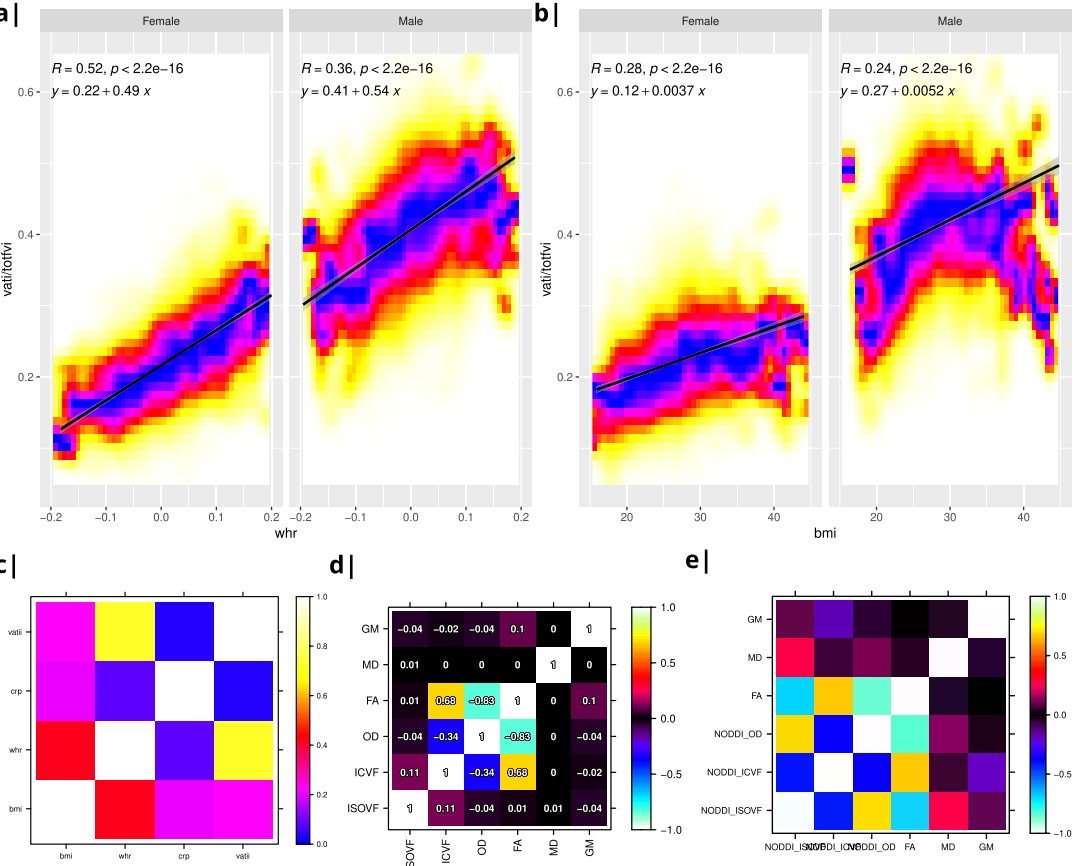

**Appendix 2—figure 1.** Internal structure of input data. Waist-to-hip ratio (WHR) in (**a**) has a much tighter linear relationship with relative visceral adipose tissue from MRI scans than BMI in (**b**). Bottom row: internal correlation in adiposity data (**c**), imaging data (**d**), and imaging-WHR maps (**e**). GM = Grey Matter; MD = Mean Diffusivity; FA = Fractional Anisotropy; OD = Orientation Dispersion Index; ISOVF = isotropic volume fraction; ICVF = intra-cellular volume fraction; BMI = body mass index; WHR = waist-to-hip ratio; CRP = C-reactive protein; VATI = visceral adipose tissue index; TOTFVI = total fat volume index.

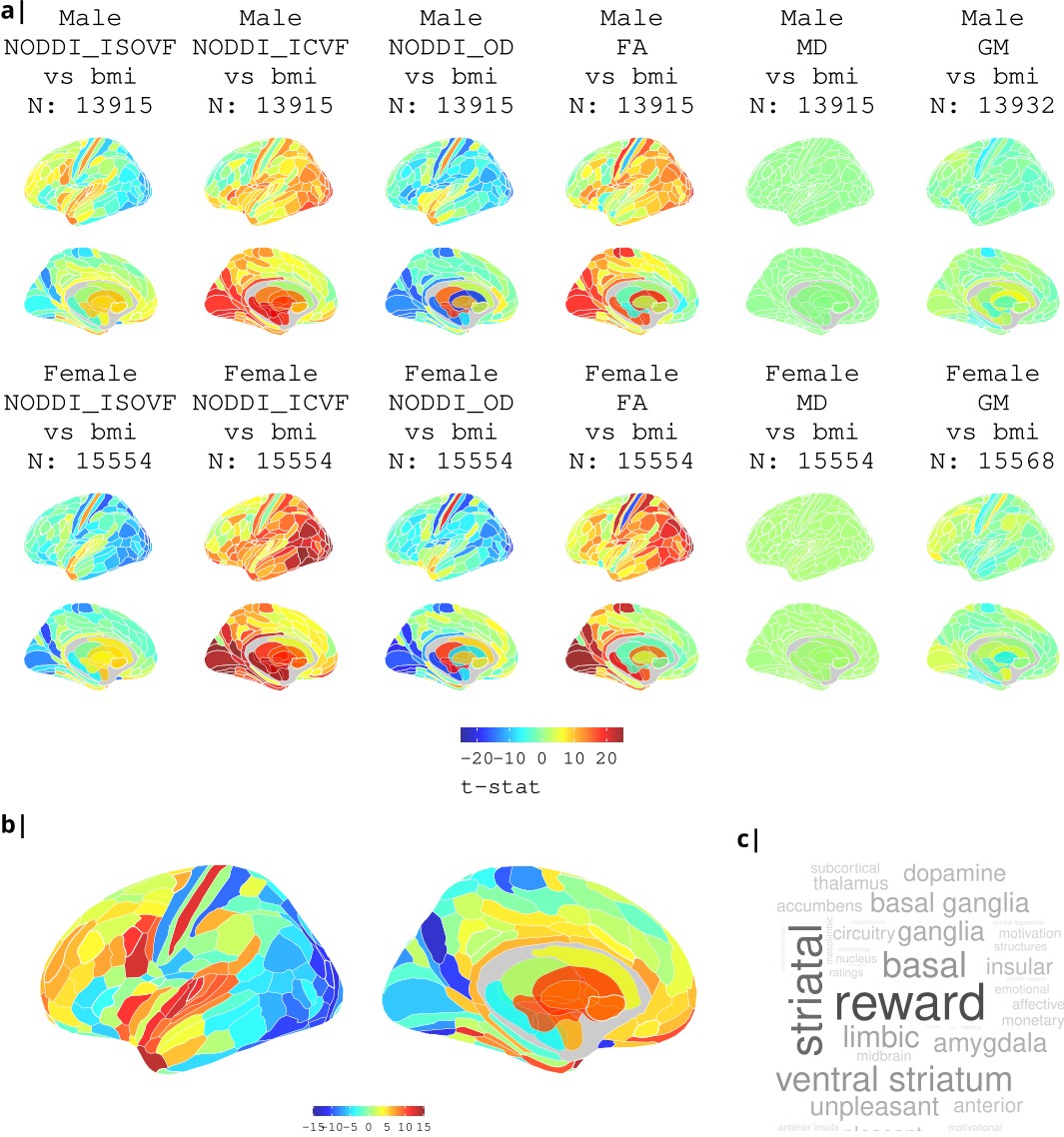

**Appendix 2—figure 2.** Association of various MRI metrics with BMI. (**a**) Brain maps showing dependence of NODDI metrics and gray matter density on body mass index, separately for males and females. Bottom: (**b**) enlarged ISOVF-BMI map and (**c**) corresponding terms from Neurosynth arranged as a word cloud.

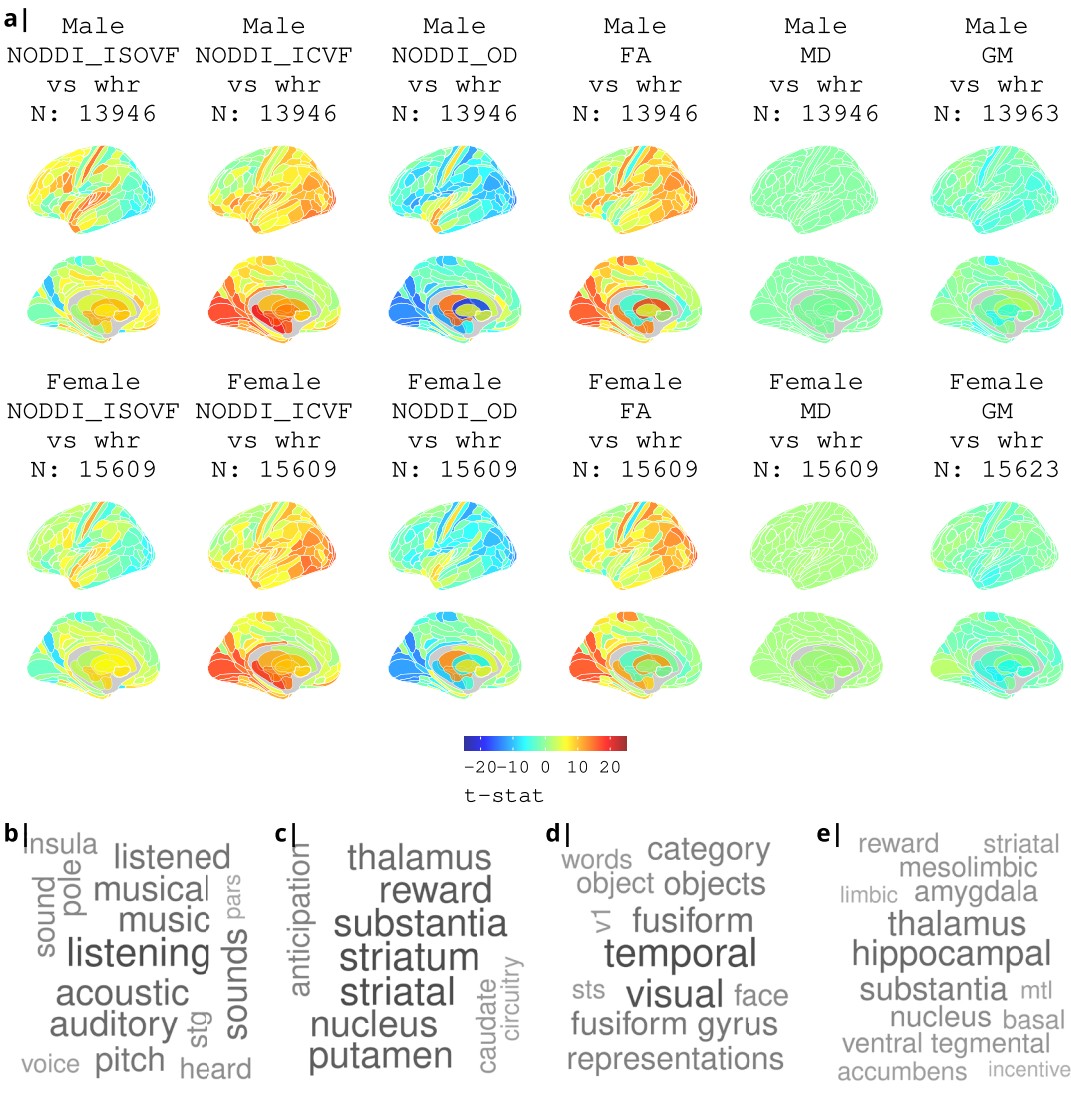

**Appendix 2—figure 3.** Association of various MRI metrics with BMI. (**a**) Brain maps showing dependence of NODDI metrics and gray matter density on body mass index, separately for males and females. Bottom: terms from Neurosynth arranged as a word cloud corresponding respectively to (**b**) ISOVF, (**c**) ISOVF sub-cortical, (**d**) ICVF, and (**e**) ICVF sub-cortical maps.

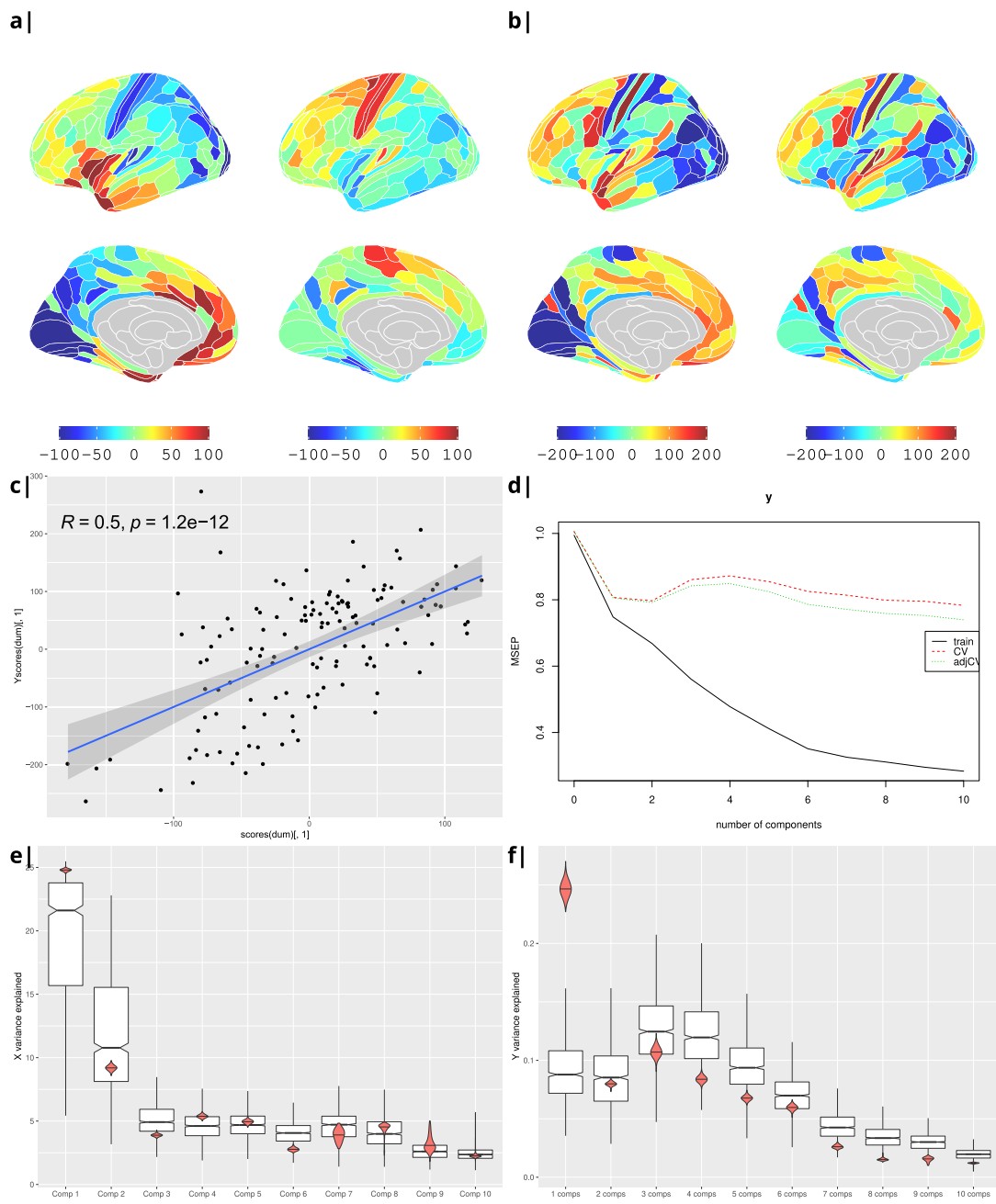

**Appendix 2—figure 4.** Body mass index. Gene correlational maps of first two X scores (**A**) and Y scores (**B**). Scatterplot of X vs Y scores across ROIs (**C**). (**D**) Cross validation of the PLS analysis. Only the first component contributes significantly to reduce the mean square error of the prediction. (**E**) and (**F**) Explained variance in X and Y respectively per component in real data (red) compared to surrogate data (boxes).

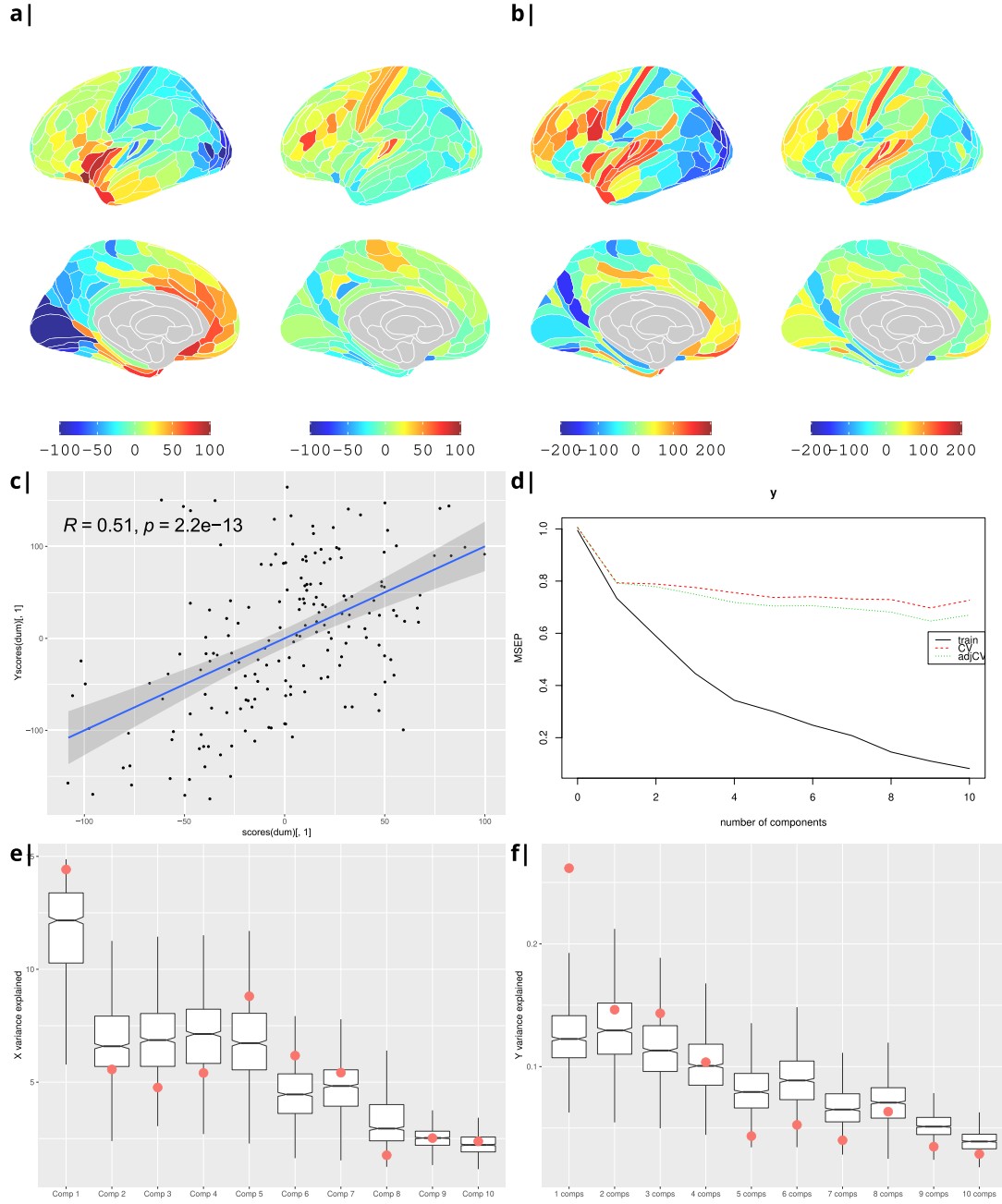

**Appendix 2—figure 5.** Waist-to-hip ratio: gene correlational maps of first two X scores (**A**) and Y scores (**B**). Scatterplot of X vs Y scores across ROIs (**C**). (**D**) Cross validation of the PLS analysis. Only the first component contributes significantly to reduce the mean square error of the prediction. (**E**) and (**F**) Explained variance in X and Y respectively per component in real data (red) compared to surrogate data (boxes).

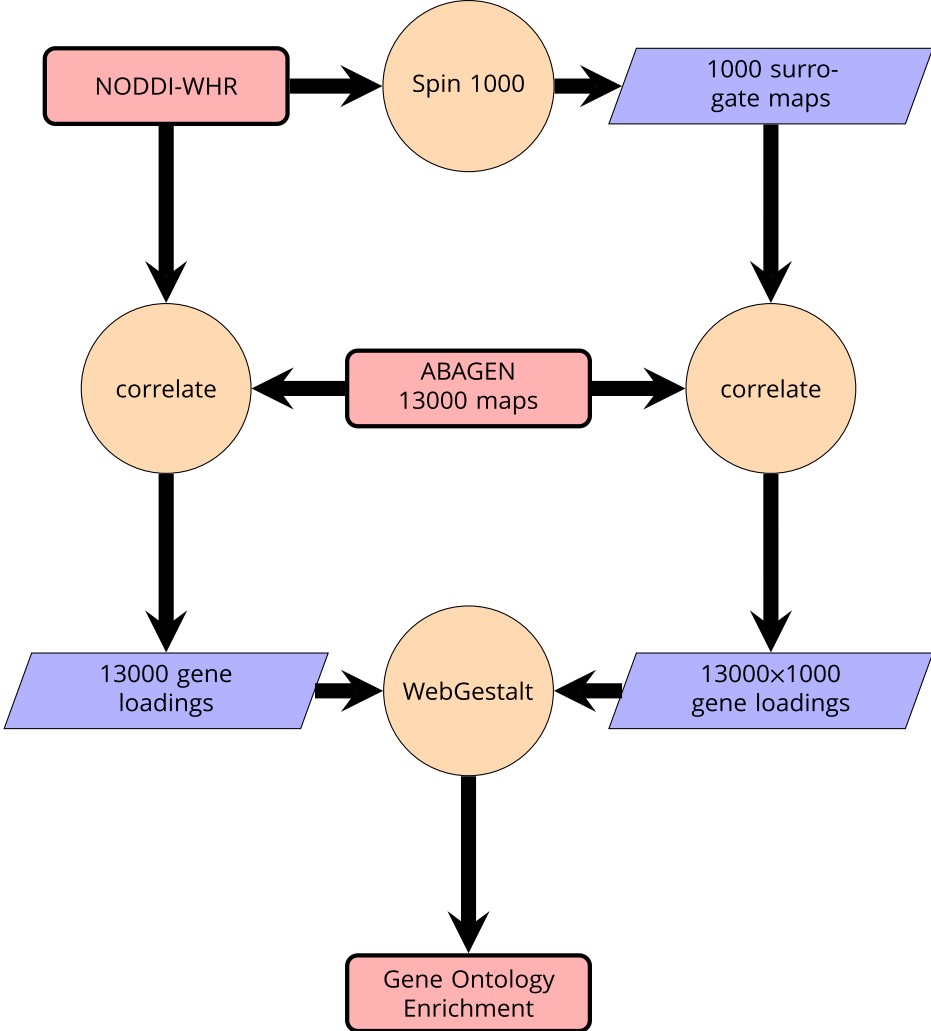

**Appendix 2—figure 6.** Schematic of analysis pipeline for gene ontology analysis with *Webgestalt* based on the correlation of NODDI-WHR and gene expression maps. Significance calculation is based on permutations taking into account the smoothness of cortical patterns (spin permutations).

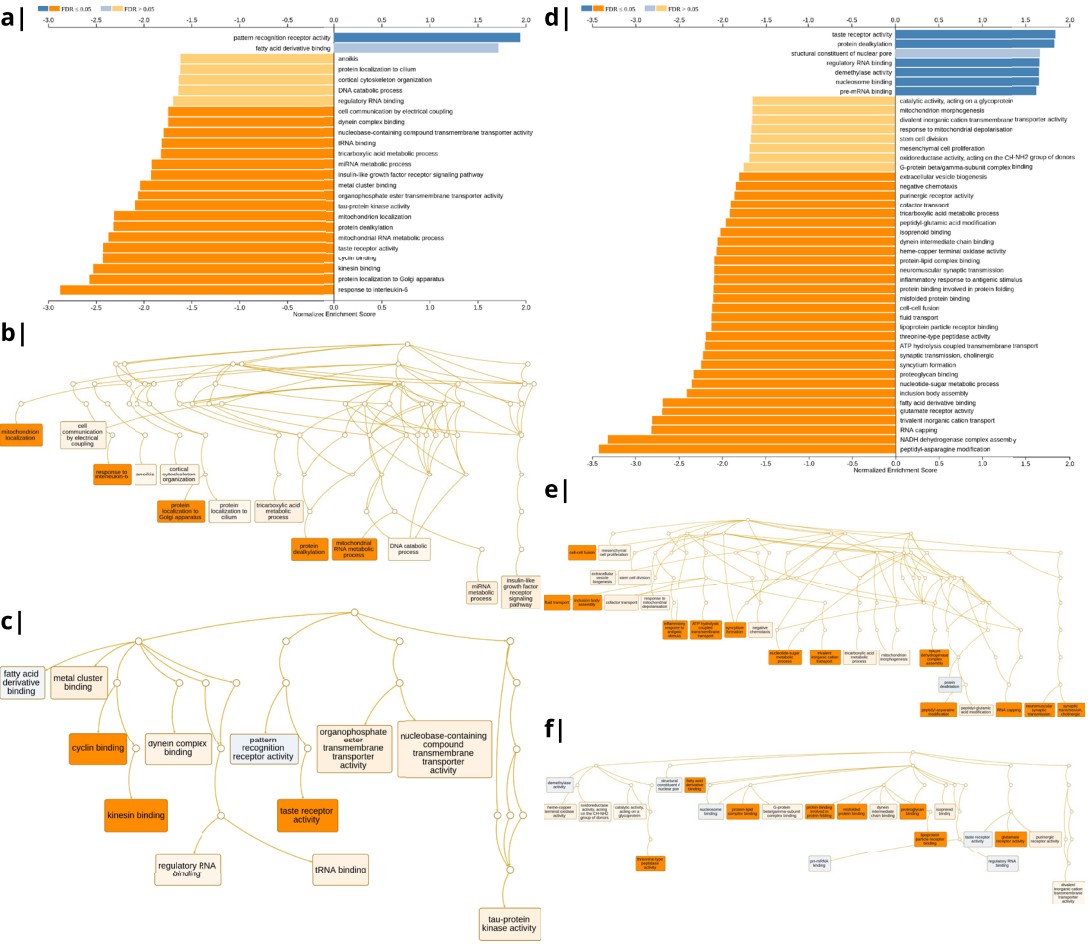

**Appendix 2—figure 7.** Waist-to-hip ratio: significantly enriched gene ontology categories according to *Webgestalt* based on the correlation of NODDI-WHR and gene expression maps. Left: results using the ISOVF-WHR maps (free water vs adiposity). (**a**) bar graph of significant gene ontologies showing normalized enrichment score on the x-axis. (**b**) Directed acyclic hierarchical graph (DAG) of GOs in the Biological Processes category. (**c**) DAG of GOs in the Molecular Function category. Right: (**d-f**) are exactly the same as (**a-c**) on the left, using instead the ICVF-WHR maps (neurite density vs adiposity). Significance calculation is based on permutations taking into account the smoothness of cortical patterns (spin permutations).

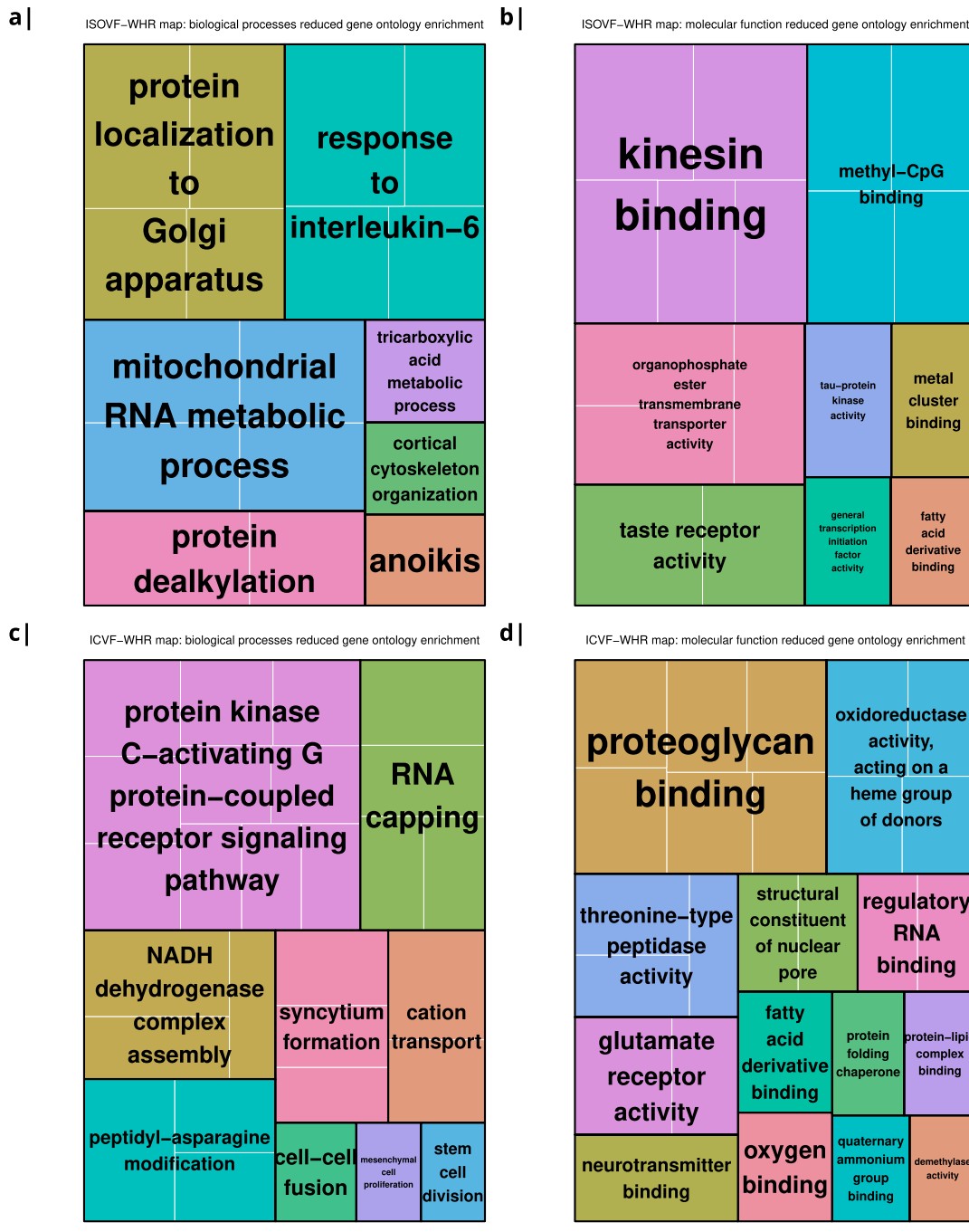

**Appendix 2—figure 8.** Significantly enriched gene ontology categories according to *Webgestalt* based on the correlation of NODDI-WHR and gene expression maps. (**a–b**) same results as in *Figure 2a* but with semantically reduced GO categories illustrating hierarchical dependencies. Results are split by category: biological processes (**a**) and molecular function (**b**). (**c–d**) same results as in *Figure 2b* but with semantically reduced GO categories split by category: biological processes (**c**) and molecular function (**d**).

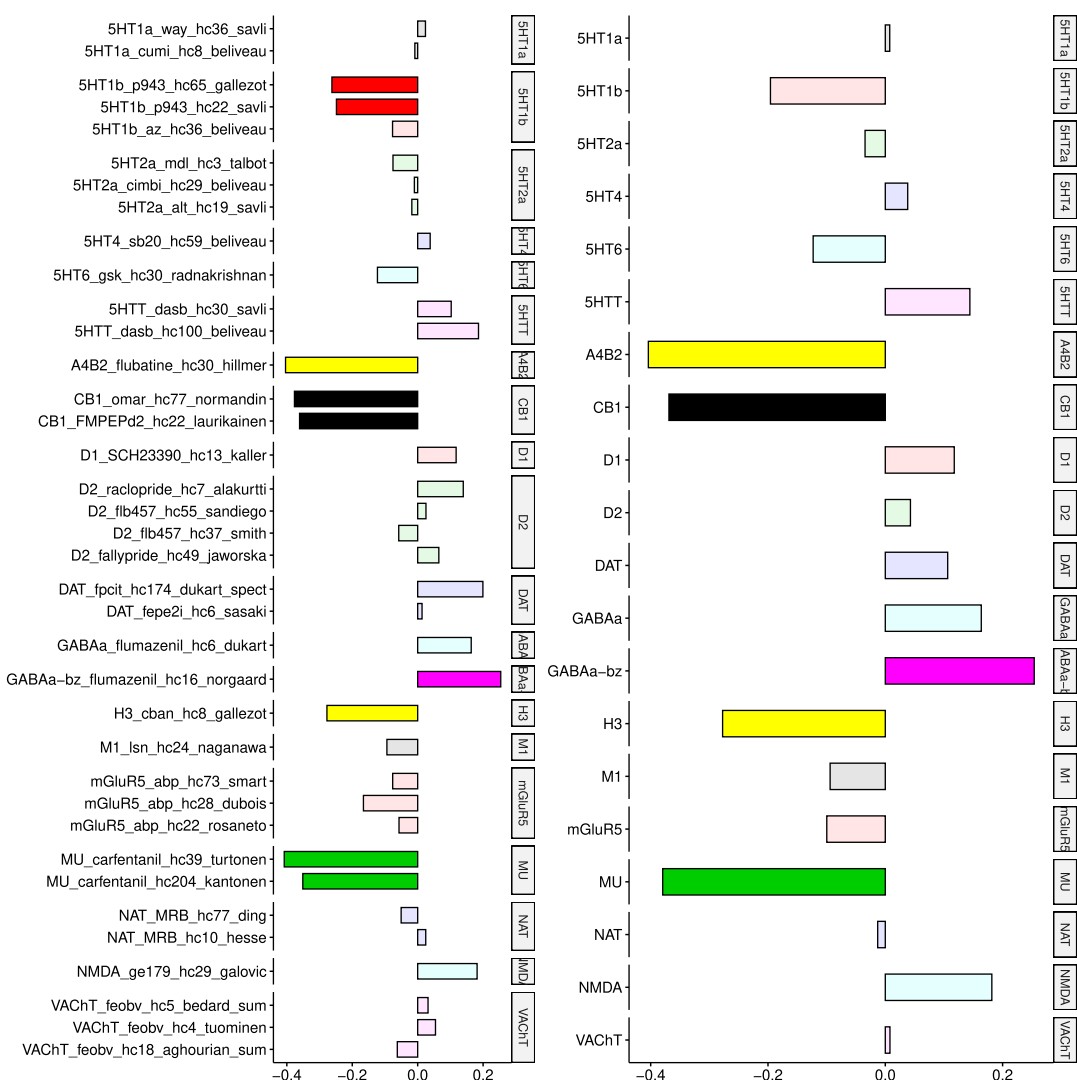

**Appendix 2—figure 9.** Body mass index: correlations of cortical neurotransmitter maps from the literature with the NODDI ICVF-BMI maps shown above. Significance after Bonferroni correction is indicated by shading. Left: individual studies, right: same neurotransmitters from different studies combined. The CB1 (cannabinoid) receptors show the maximum (absolute) correlation with the maps of microstructural effect of obesity.

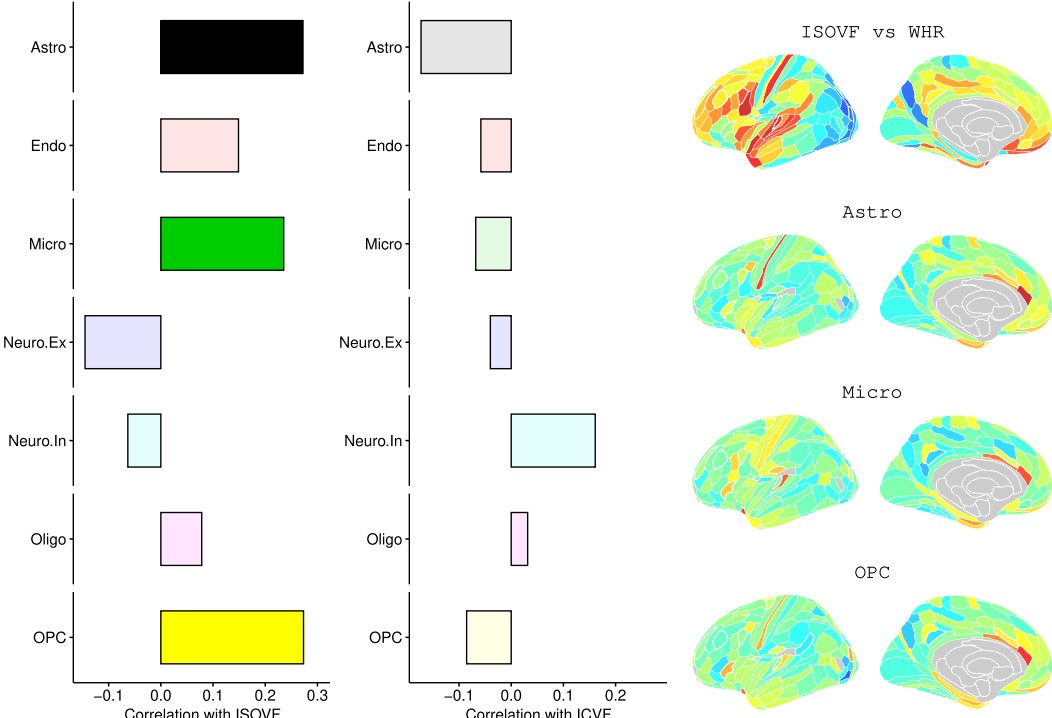

**Appendix 2—figure 10.** Left: Correlations of Brain cell type maps for 31 cell types from *Lake et al., 2018* with the NODDI ISOVF and ICVF-WHR maps shown above. Significance is indicated by shading (based on spin permutation and Bonferroni correction). Right: The Astrocytes, Microglia, and OPC cell type maps show the maximum (absolute) correlation with the ISOVF maps of microstructural effect of obesity.

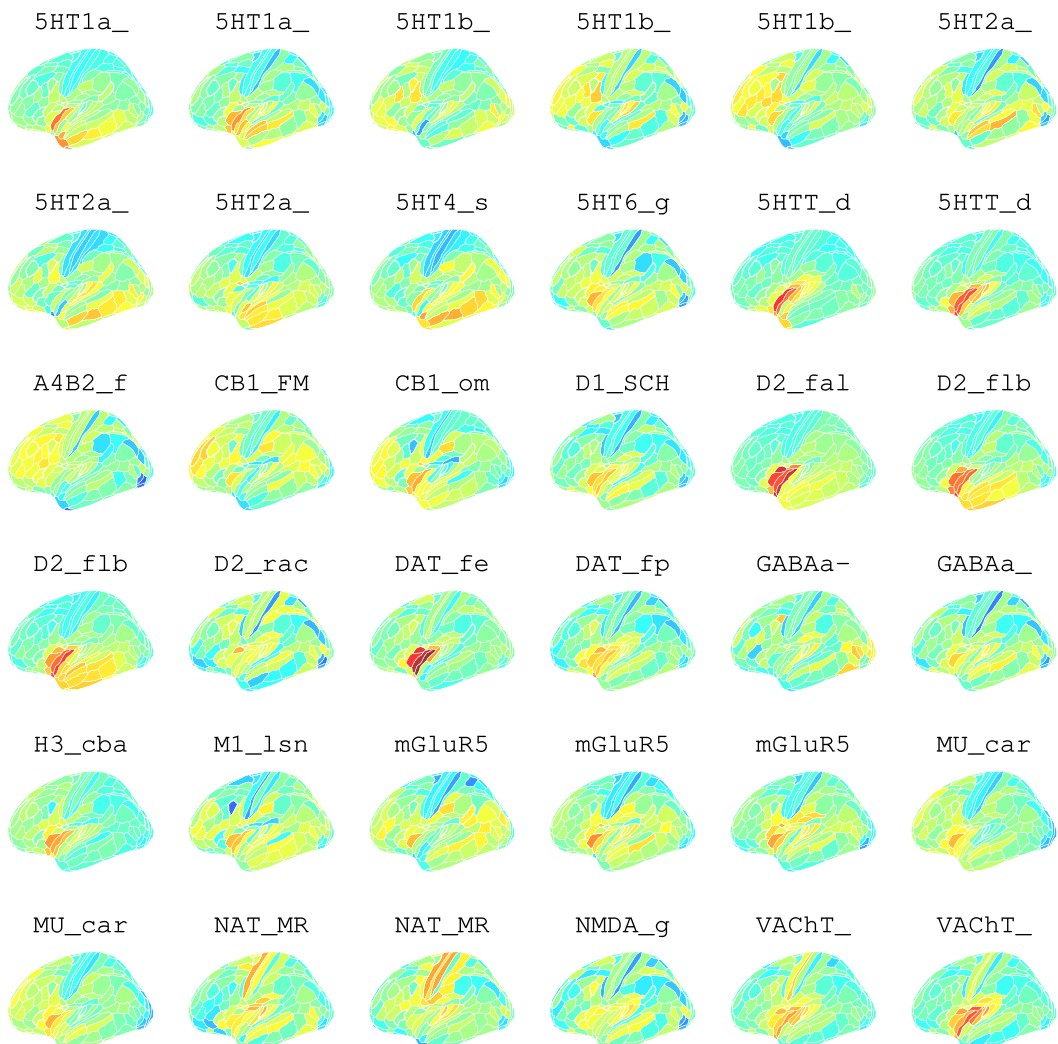

**Appendix 2—figure 11.** Neurotransmitter maps for 36 neurotransmitters from *Hansen et al., 2022*.

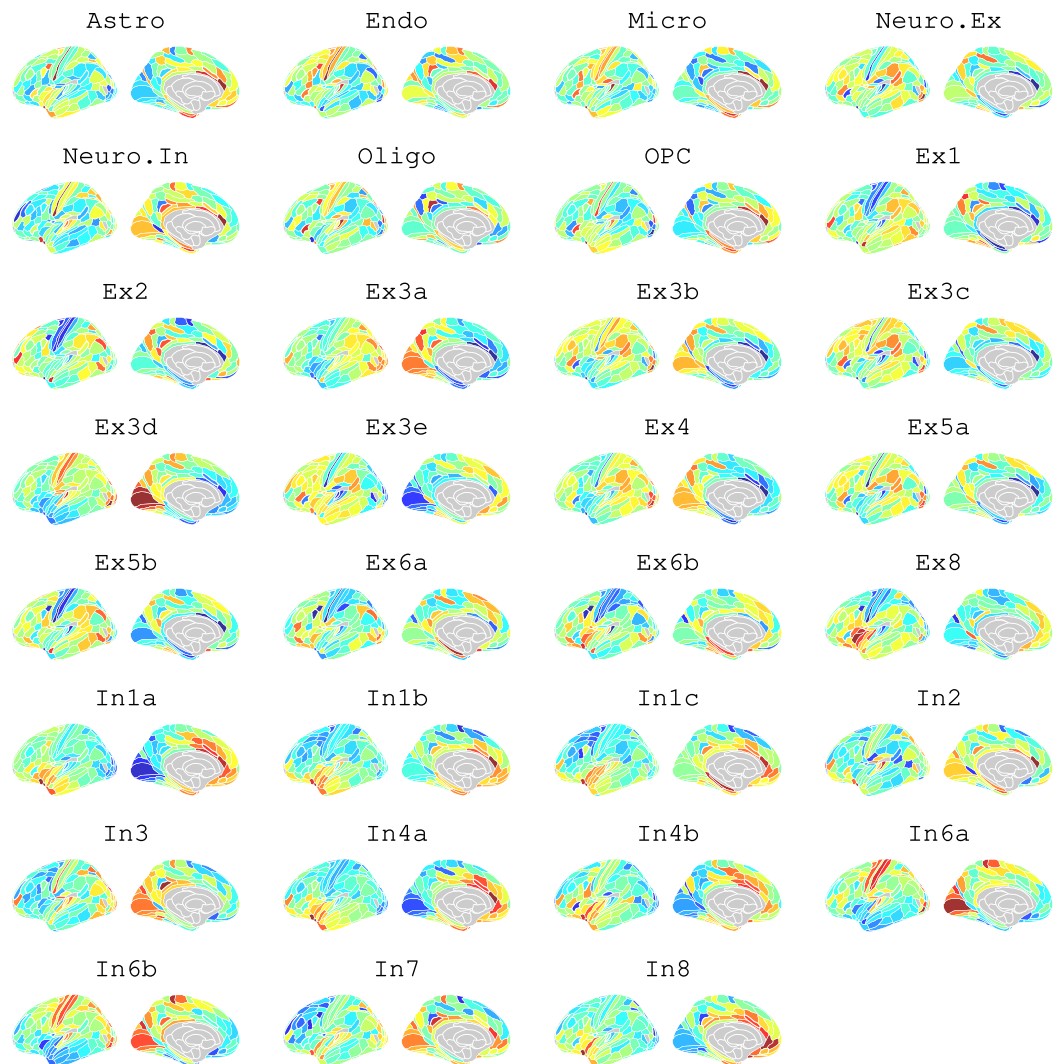

**Appendix 2—figure 12.** Brain cell type maps for 31 cell types from *Lake et al., 2018*.

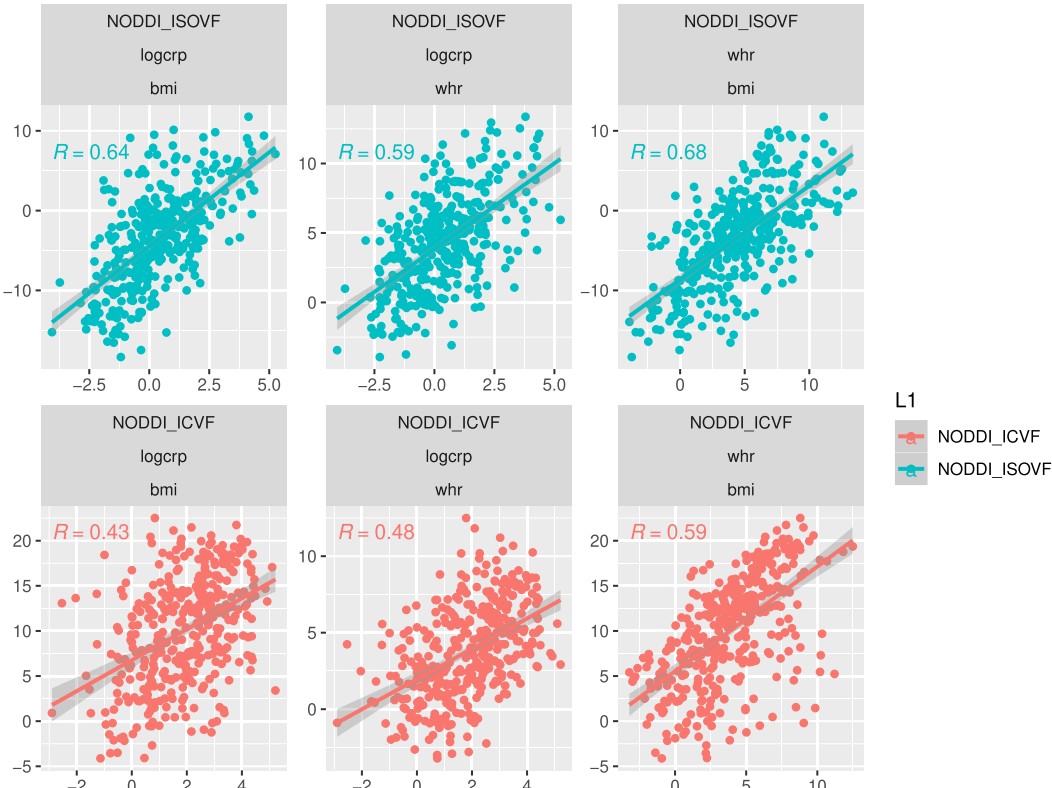

**Appendix 2—figure 13.** Scatterplot over 376 regions of pairwise relationships between t-score maps for variables WHR, BMI, and CRP respectively. Top: similarity between ISOVF maps, bottom: similarity between ICVF maps. Calculating statistics based on Fisher transformed correlation values, for both CRP pairs the correlation is significantly stronger for the ISOVF maps than the ICVF maps (CRP-BMI: $P < 1.2 \times 10^{-5}$, CRP-WHR: $P < 0.024$, one-tailed) and we also find that the BMI and WHR maps are marginally different (BMI-WHR: $P < 0.05$, two-tailed).

