## [Editor Report]

Kitzbichler et al. conducted a valuable large-scale study using the UK Biobank data to explore the relationship between brain tissue microstructure and obesity and provided convincing evidence for two coupled yet distinct brain systems mediating relationships between free water and neurite density as markers of inflammation with the genes enriched for innate immunity and specific neurotransmitter receptors. Major strengths include the innovative and expansive approach to understanding the genetic factors, neurotransmitters and potential mechanisms underlying observed alterations in cortical thickness and gray matter volume in obesity. The scope of the work goes beyond most standard neuroimaging analyses and reveals coherent patterns linking neurite density and free water to relevant neuroinflammatory and neurotransmitter pathways.

---

## [Decision Letter]

**Decision letter after peer review:**

[Editors’ note: the authors submitted for reconsideration following the decision after peer review. What follows is the decision letter after the first round of review.]

Thank you for submitting the paper "Two differentiated brain systems micro-structurally associated with obesity" for consideration by *eLife*. Your article has been reviewed by 2 peer reviewers, and the evaluation has been overseen by a Reviewing Editor and a Senior Editor. The following individual involved in the review of your submission has agreed to reveal their identity: Annette Horstmann (Reviewer #1).

Comments to the Authors:

We are sorry to say that, after consultation with the reviewers, we have decided that this work will not be considered further for publication by *eLife*.

I am sorry to convey this disappointing news. This is clearly an interesting work. As you will see below, the reviewers had mixed opinions, with one reviewer considering the work to be an important advance with some methodological limitations, and the other finding the approach to be too exploratory with major methodological flows. Given the reservations regarding the methodology, the article does not represent an advance with sufficient rigor to merit publication in *eLife*.

*Reviewer #1 (Recommendations for the authors):*

The study uses a large and well-suited human dataset to investigate the relationship between brain microstructure, as determined by in vivo MRI, and obesity, as measured by waist-to-hip ratio, in humans. The authors identify two brain systems that are individually associated with two different markers of brain structure: Neurite density (ICVF) and free water (ISOVF), a potential indicator of inflammatory processes. The authors link the identified associations to biological processes and in particular to neurotransmitter systems using publicly available expression atlases. This is a definite strength as it paves the way to a deeper and mechanistic understanding of the observed associations. Finally, they use GWAS analyses to investigate the influence of genetic variation on the associations between brain microstructure and obesity. They find that only one of the associations is linked to genetic variation and conclude that the structural differences in the two brain systems could be interpreted as a cause or consequence of obesity, respectively.

The analyses reported in the main manuscript are limited to the waist-to-hip ratio. This measure of obesity is correlated with BMI but primarily captures visceral obesity. The complementary results suggest that the results obtained using BMI are much more influenced by gender. The motivation to look at one and not the other is not well established. The same applies to the choice of MRI markers: the decision to use neurite density and free water as the main dependent variables is based on the strength of their association with the predictors. It is not clear from the data whether the associations with the two brain systems are present in the same individuals or whether these results come from two different subgroups within the data set. The authors do not support their interpretation with further (potentially available) data, e.g. on inflammatory status.

Overall, the data are of great interest and the technical methods are sound. Some methodological choices and selective reporting of results weaken the validity of the conclusions at this stage.

– The sole focus on WHR is not well justified. Although BMI and WHR share a large proportion of the variance, the correlation is only r = 0.428. Furthermore, the supplementary figures show that the association between MRI and BMI varies much more by gender than WHR. Please point out in the manuscript that WHR is primarily a measure of central, i.e. visceral obesity, which is metabolically less favourable but also has gender differences, and discuss the choice of predictor more thoroughly. It would be even more interesting if both measures could be analysed, as WHR may be more closely associated with low-grade inflammation than BMI.

– The motivation to study the specific measures of microstructure is insufficiently justified. The stated motivation does not go far beyond stating that "no one has done this before" and that these metrics were most strongly correlated with WHR, i.e. 'To address this potential redundancy, we performed a preliminary correlational analysis of all 6 MRI metrics with WHR then focused our subsequent analyses on ICVF and ISOVF, the two complementary MRI metrics that were most strongly associated with WHR.' Please explain in more detail what these metrics contribute in addition to the other metrics reported in the literature.

– Are the 'distinct brain phenotypes' present in the same individuals or do they represent different subgroups?

– What does it mean that 'The maps of ISOVF~WHR and ICVF~WHR were negatively correlated with r = -0:366'? This indicates that these systems (or the underlying measures/analyses) are not independent. How can this be explained? Does this not weaken the interpretation of differential association?

– Why was GM but not cortical thickness included in the MRI measurements? This would be informative as results have already been published on the relationship between cortical thickness and obesity.

– What does it mean that "taste receptor activity is under-represented in the enrichment analysis for one but positively enriched in the analysis for the other MRI marker? If both are valid findings, please report both and not selectively report one.

– Have inflammatory markers been investigated? For example, if IL-6 was associated with MRI markers in the same way as WHR, this would support the authors' interpretation of the data.

*Reviewer #2 (Recommendations for the authors):*

Strengths

The article takes an expansive, whole-brain approach to addressing the question of interest. While the methodology is simple (i.e., correlations), the rigor of the statistical method ensures that some of the results are plausible and are likely to be replicated. While I disagree with the range of techniques used, there is a comprehensive quality to them which for some reviewers and readers will be impressive.

Finally, the figures are all high-quality and very well made, though incorrect or overreaching elements do cloud this otherwise rosy picture.

Weaknesses

In the introduction, the authors take a straightforward approach to address what is terrible literature, as I describe at length in a previous article. There are inconsistencies in how the brain is assessed, measures that are used to gauge obesity, the wide variance in sample size, poor statistical analysis of many brain regions, and so on. It is much less "pat" than the authors describe, and they really need to spend more time describing the non-significant and inconsistent findings over several brain regions described. The transition to discussing DTI is also incomplete. Finally, I think it is odd to examine 180 cortical regions in a systematic fashion, rather than limiting analyses to larger regions and focusing on smaller regions based on initial analyses. This approach is faulty and prone to error, one which is exacerbated by FDR which looks great on paper but in practice is highly prone to type 1 error given enough comparisons.

For methods, describe the final sample size in the main article. Do not relegate it to supplemental, as the sample size is really around 31,500 instead of 34,000 for most indices. I do not understand why people with high CRP values were excluded, yet scans were kept with participants that had all-cause dementia, heart failure, stroke, and a variety of other conditions that reliably show differences in the frontal lobe and other volumes. Only one scanner is described for UK Biobank but this is not correct. Three scanners were used, as described later on in the analysis pipeline. While the site can be added as a covariate, some studies looking at the effect of scanner units have been done and should be included. The use of the Allen Brain Atlas for gene expression data is unwise. Last I checked, a literal few participants in the same age range as the UK Biobank participants had been scanned using this technique.

For results, BMI is not a good representation of central obesity and should be ignored. That it is relegated to supplemental is wise. WHR is better, but a correlation of r=0.428 is not "strong." It is a small-moderate effect size and should be treated appropriately. I find the use of volumetrics, DTI, and DTI-NODDI to be strange. For each region, technically, there should be 4 standard DTI metrics and 3 NODDI metrics.

Figure 1 is slick. I do not see the point of having a correlation matrix like Figure 1a. For Figures 1b and 1c, as well as Figures 1d and 1e, what I am struck by is the small correlation found in the areas that show the strongest association. That higher WHR is related to greater ICVF in the hippocampus is important, for example, but a correlation of 0.32 is hardly something to worry about. I demand a range of correlation matrices besides what looks best. Further, I find the associations with the occipital lobe hard to believe. This region rarely shows significant associations with obesity. Likewise, I find the frontal and in particular prefrontal areas showing no association to be surprising. In comparing all participants aged 40-70 years, perhaps comparisons by binned age groups would be better.

I do not see the point of using the Allen Brain Atlas, as described. It is not only non-representative of gene expression given its very small sample size, but a lack of suitability to the British population versus the handful of younger to a literal few older Americans used to make the maps. Use Bonferroni correction to truly take into account the number of comparisons made and get back to me. What I most strongly object to is the lack of actual metrics (i.e., r, R^2, Cohen's D, etc.) describing any of the correlations. We are shown plots but are not given correlations in many cases (e.g., Figure 4). This makes for more slick figures, but to what end and for what purpose? I am not going to win over by glam and type-setting.

The discussion is off-putting. To be clear, the authors focused on two DTI-NODDI indices and relegated three other metrics to supplemental because they were less impressive. This is not a hypothesis. There was no a priori determination that ISOVF and ICVF would show impressive results. This was the result of an FDR-corrected smorgasbord of analyses that the authors hoped would win over reviewers because of how nice everything looked. I do machine learning and limited computational biology techniques every day. I am not impressed. The rest of the discussion is based on a corpus of findings that are either not appropriate (e.g., Allen Brain Atlas) or many series of FDR-corrected findings that I do not believe.

[Editors’ note: what follows is the reviewers’ response to the second round of review.]

*Reviewer #3 (Recommendations for the authors):*

Kitzbichler et al. performed a large-scale study using UK Biobank data to explore the relationship between brain tissue microstructure and obesity. The authors uncover two coupled yet distinct brain systems mediating relationships between free water and neurite density as markers of inflammation with the genes enriched for innate immunity and specific neurotransmitter receptors. This paper offers an innovative and expansive approach to understanding the genetic factors, neurotransmitters, and potential mechanisms underlying observed alterations in cortical thickness and gray matter volume in obesity. The scope of the work goes well beyond most neuroimaging analyses and reveals coherent patterns linking neurite density and free water to relevant neuroinflammatory and neurotransmitter pathways.

Strengths of the paper include the use of a large and well-substantiated human dataset alongside rigorous, albeit limited, gene expression data derived from the Allen Brain Atlas. The GWAS analyses are well integrated and provide insights into the impact of genetic variation on associations between gray matter microstructure and obesity, thereby offering a more comprehensive understanding of the observed relationships. While it would be helpful to include additional data to support the interpretations, the manuscript presents a sufficiently comprehensive body of evidence to indicate the presence of distinct brain systems mediating the effect of obesity on brain tissue microstructure and macrostructure.

The results support an intriguing mechanistic model in which neurite density in the medial temporal-occipital-striatal system may contribute to obesity, while free water in the prefrontal-temporal striatal system may reflect the consequences of obesity mediated by innate immune system function. The findings provide evidence that microstructural imaging measures derived from diffusion MRI may offer more specific insights into the biological mechanisms underlying previously observed alterations in cortical thickness and gray matter volume in obesity. The work offers a notable advance in understanding the link between obesity, neuroinflammation, and alterations in brain tissue microscopic structure and provides a framework for deriving further mechanistic insights through this approach that may be applied to other neurological diseases and systemic disorders.

The authors have addressed the critiques from the previous round of review thoroughly. All major criticisms have been mitigated by additional data and appropriate responses. While the critiques from the second reviewer were phrased strongly, the authors have done their best to address the most salient comments objectively. As a result, the revisions to the manuscript have greatly strengthened its presentation and provide a more comprehensive and well-reasoned paper.

---

## [Author Response]

[Editors’ note: The authors appealed the original decision. What follows is the authors’ response to the first round of review.]

Reviewer #1 (Recommendations for the authors):1. The study uses a large and well-suited human dataset to investigate the relationship between brain microstructure, as determined by in vivo MRI, and obesity, as measured by waist-to-hip ratio, in humans. The authors identify two brain systems that are individually associated with two different markers of brain structure: Neurite density (ICVF) and free water (ISOVF), a potential indicator of inflammatory processes. The authors link the identified associations to biological processes and in particular to neurotransmitter systems using publicly available expression atlases. This is a definite strength as it paves the way to a deeper and mechanistic understanding of the observed associations.

We thank the reviewer for their positive comments about the strengths of our work.

2. Finally, they use GWAS analyses to investigate the influence of genetic variation on the associations between brain microstructure and obesity. They find that only one of the associations is linked to genetic variation and conclude that the structural differences in the two brain systems could be interpreted as a cause or consequence of obesity, respectively.The analyses reported in the main manuscript are limited to the waist-to-hip ratio. This measure of obesity is correlated with BMI but primarily captures visceral obesity. The complementary results suggest that the results obtained using BMI are much more influenced by gender. The motivation to look at one and not the other is not well established.

Thank you for raising this point. Our primary motivation for focussing on WHR was because of its greater association with visceral obesity (as also highlighted by R2). Nevertheless, we performed (and reported) all analyses for both WHR and BMI. For the most part, findings for WHR and BMI were broadly comparable, as illustrated in Figure S13 which explicitly compares the BMI and WHR maps. For clarity, we presented the WHR results in the main text and placed all equivalent results for BMI in the SI. We have now clarified this decision to focus on WHR in the main text as highlighted in our detailed replies below.

3. The same applies to the choice of MRI markers: the decision to use neurite density and free water as the main dependent variables is based on the strength of their association with the predictors.

We apologise for not being clearer in our rationale for focussing on neurite density and free water. As we have now further clarified in the text, our goal was to use NODDI modelling of diffusion MRI to investigate associations of obesity with brain microstructure. We focussed on NODDI as this provides three (reasonably well validated and interpretable) indices of tissue microstructure. Of note, standard diffusion metrics such as mean diffusivity and fractional anisotropy of grey matter have poor interpretability with respect to cortical microstructure and were only included to illustrate their relationship to the NODDI metrics.

Our decision to restrict ourselves to two of the three NODDI metrics (i.e. excluding OD) was largely driven by a desire for brevity. The ICVF and OD maps were strongly correlated, so we elected to illustrate results for the more independent and complementary measures of ISOVF and ICVF in the main text and report the results on OD in SI rather than reporting the OD results in parallel to the closely related ICVF results in the main paper.

In response to the reviewer’s comments, we have now repeated the analysis using PCA components rather than raw NODDI measures. This confirmed that ISOVF was an independent measure and that ICVF and OD were co-linear. We have now clarified this in the paper as copied below. However, we stuck with our decision to report NODDI metrics (rather than PCA components) as this is generally more easily interpretable for the general reader.

The detailed response below discusses again in more detail these questions and also shows the textual changes made.

4. It is not clear from the data whether the associations with the two brain systems are present in the same individuals or whether these results come from two different subgroups within the data set.

This is an interesting question. Our assumption is that both systems would be in operation in the same individuals. However, we are not aware of any currently available tools that would allow us to undertake this analysis within a single subject.

5. The authors do not support their interpretation with further (potentially available) data, e.g. on inflammatory status.

Thank you for raising this interesting point. UK Biobank does not provide data on IL or other proinflammatory cytokines but it does provide CRP data as a broad index of systemic inflammation. As suggested we have now undertaken an additional analysis of associations between CRP and the microstructural MRI metrics to test the hypothesis that brain maps relating to CRP will be more similar to (more strongly correlated with) maps of obesity scaling with ISOVF (which we hypothesised represent potentially inflammatory effects of obesity on the brain) than to maps of obesity scaling with ICVF (which we hypothesised represent potentially causal effects of brain-mediated behaviour on obesity).

We have now addressed this point in the results and Discussion sections (as well as the abstract) and have added a new figure (Figure S13, also included in the next section for reference) which shows two main findings: (1) that there is a moderately strong association of CRP with microstructural brain metrics similar but weaker than WHR or BMI; and (2) that this relationship was statistically significantly stronger for the effect maps of CRP vs ISOVF compared to the maps of CRP vs ICVF as would be expected if the free water measured by ISOVF is related to inflammatory effects of obesity on the brain.

6. Overall, the data are of great interest and the technical methods are sound. Some methodological choices and selective reporting of results weaken the validity of the conclusions at this stage.

We thank the reviewer for their positive assessment of the relevance of our work. We hope that we have been able to better clarify the motivation for our methodology and show that this has been driven by a desire for clarity rather than a selective focus on reporting specific findings (all of which are reported in SI). We also hope that our additional analyses have provided further support for our conclusions.

7. The sole focus on WHR is not well justified. Although BMI and WHR share a large proportion of the variance, the correlation is only r = 0.428. Furthermore, the supplementary figures show that the association between MRI and BMI varies much more by gender than WHR. Please point out in the manuscript that WHR is primarily a measure of central, i.e. visceral obesity, which is metabolically less favourable but also has gender differences, and discuss the choice of predictor more thoroughly. It would be even more interesting if both measures could be analysed, as WHR may be more closely associated with low-grade inflammation than BMI.

We have performed all analyses in parallel for both WHR and BMI but decided for the sake of clarity to present only the WHR results in the main text and provide the BMI results in the SI for the interested reader. Additional text justifying our preference for WHR as the principal marker of (visceral) obesity and discussing the metabolic importance of visceral fat has been added to the main text:

“To date, cross-sectional and longitudinal studies investigating effects of obesity on the brain have focused almost exclusively on macroscopic aspects of brain structure such as total grey matter volume and cortical thickness. Typically, these have reported negative associations between obesity (particularly visceral obesity indexed by waist to hip ratio: WHR) and (smaller) total grey matter volume (Cox et al., 2019) and (thinner) cortical thickness (Caunca et al., 2019).

[…]

Given previous findings of a particular association between macroscopic differences in brain structure and visceral obesity we elected to report associations with WHR in the main text and report complementary results for BMI as a measure of whole body obesity in the SI.”

Furthermore, the relationships between MRI-derived visceral fat and BMI as well as WHR are now presented in Figure S1 a-b, separately for males and females, to make it clear that this effect is not gender-driven and to demonstrate that the relationship with MRI measures of visceral fat is more strongly linear for WHR compared to BMI.

Concerning the microstructural effects, findings for WHR and BMI were broadly comparable, as can be also seen in Figure S13 which explicitly compares the BMI and WHR maps.

8. The motivation to study the specific measures of microstructure is insufficiently justified. The stated motivation does not go far beyond stating that "no one has done this before" and that these metrics were most strongly correlated with WHR, i.e. 'To address this potential redundancy, we performed a preliminary correlational analysis of all 6 MRI metrics with WHR then focused our subsequent analyses on ICVF and ISOVF, the two complementary MRI metrics that were most strongly associated with WHR.' Please explain in more detail what these metrics contribute in addition to the other metrics reported in the literature.

We apologise for not being clearer in our rationale for focussing on neurite density and free water. We note that the reviewer made a similar comment (R1, #3) previously and our response to this comment substantially reproduces our response to R1, #3. As we have now further clarified in the text, our goal was to use NODDI modelling of diffusion MRI to investigate associations of obesity with brain microstructure. We focussed on NODDI as this provides three (reasonably well validated and interpretable) indices of tissue microstructure. Of note, standard diffusion metrics such as mean diffusivity and fractional anisotropy grey matter have poor interpretability with respect to cortical microstructure and were only included to illustrate their relationship to the NODDI metrics.

We have rewritten the text to better motivate our choice of microstructural metrics and why these offer a new and complementary approach as copied below:

“However, changes in grey matter volume and cortical thickness can be driven by multiple different underlying processes and our understanding of the microstructural features that underpin this relationship remain largely unknown (Westwater et al., 2022). For example, it is currently not known whether obesity-associated differences in grey matter volume relate to changes in the size, shape or number of neurons e.g. neurite density or orientation dispersion within that region or alternately to differences in tissue water content.

[…]

Unlike conventional diffusion MRI which models data acquired at a single diffusion weighting (shell), NODDI requires data collected at multiple different diffusion weightings (shells) then exploits the diffusion characteristics that can be observed in different tissue compartments to quantify their respective volume fractions. In this model, diffusion is modelled as isotropic in free water, restricted within neurites, and hindered in the extracellular space resulting in three microstructural metrics: Intracellular Volume Fraction (ICVF) which captures the volume fraction occupied by neurites (axons and dendrites) but not cell bodies, Orientation Dispersion Index (OD) which captures the spatial distribution of these processes and isotropic volume fraction (ISOVF) which provides a measure of free water index).”

Our decision to restrict ourselves to two of the three NODDI metrics (i.e. excluding OD) was largely driven by a desire for brevity. The ICVF and OD maps were strongly correlated, so we elected to illustrate results for the more independent and complementary measures of ISOVF and ICVF in the main text and report the results on OD in SI rather than reporting the OD results in parallel to the closely related ICVF results in the main paper.

In response to the reviewer’s comments we have now repeated the analysis using PCA components rather than raw NODDI measures. This confirmed that ISOVF was an independent measure and that ICVF and OD were co-linear. We have now clarified this in the paper as copied below. However, we stuck with our decision to report NODDI metrics (rather than PCA components) as this is generally more easily interpretable for the general reader. The respective section in the main text is quoted below:

“As illustrated in Figure 1a, some of these metrics were strongly correlated, indicating that they represented similar aspects of the underlying cortical micro-structure or tissue composition. For example, FA, OD and ICVF metrics of neurite density were more strongly correlated with each other than with ISOVF, which is typically interpreted as a marker of tissue free water rather than cytoarchitectonics (Kamiya et al., 2020).

To address this potential redundancy, we performed a preliminary correlational analysis of all MRI metrics with WHR, then focused our subsequent analyses on ICVF and ISOVF, the two complementary MRI metrics that were most strongly associated with WHR. Comparable results for the other 4 metrics are reported in the Supplemental Information Figure S2.”

9. Are the 'distinct brain phenotypes' present in the same individuals or do they represent different subgroups?

We note that this comment is identical to the one raised as R1, #4 and we reproduce our response to the earlier comment here. This is an interesting question. Our assumption is that both systems would be in operation in the same individuals. However, we are not aware of any currently available tools that would allow us to undertake this analysis within a single subject. We have added some discussion of this point to the main text:

“Concerning the question whether both brain systems are in operation in the same individual at the same time, we are not aware of any currently available tools that would allow us to actually test this assumption, but it could be an interesting avenue for future work.”

10. What does it mean that 'The maps of ISOVF~WHR and ICVF~WHR were negatively correlated with r = -0:366'? This indicates that these systems (or the underlying measures/analyses) are not independent. How can this be explained? Does this not weaken the interpretation of differential association?

We agree that a shared variance explained of R^2^ ~ 13% indicates that these two systems are not entirely independent. We have now modified the title of the manuscript to de-emphasise the claim that these two systems are entirely independent or differentiated with respect to each other. However, we note that the two brain systems associated with ISOVF and ICVF were differentiated anatomically and in terms of their colocation with gene expression profiles and neurotransmitter receptor maps (as per Table 1).

As mentioned above, we have also repeated the analysis with PCA components of NODDI metrics, which are independent (orthogonal) by construction, but we found the same moderate relation in the resulting effect maps, indicating that this is probably mediated by effects of obesity.

11. Why was GM but not cortical thickness included in the MRI measurements? This would be informative as results have already been published on the relationship between cortical thickness and obesity.

The primary aim of the current manuscript was to focus on microstructural associations with obesity not to repeat macrostructural analyses previously reported by other groups (including in UK Biobank). Our decision to include GM and other macrostructural measures in Figure 1 was purely for the sake of completeness, to allow the reader to compare the novel microstructural features that we report here with macrostructural features previously reported and show that our microstructural data are indeed complementary as we have attempted to clarify in the respective section of the main text:

“Of the MRI metrics, there was one macro-structural measure (GM, grey matter volume) and micro-structural measures (MD, mean diffusivity; FA, fractional anisotropy; OD, orientation dispersion; ICVF, intra-cellular volume fraction; and ISOVF, isotropic volume fraction). As illustrated in Figure 1a, some of these metrics were strongly correlated, indicating that they represented similar aspects of the underlying cortical micro-structure or tissue composition. For example, FA, OD and ICVF metrics of neurite density were more strongly correlated with each other than with ISOVF, which is typically interpreted as a marker of tissue free water rather than cytoarchitectonics (Kamiya et al., 2020).

To address this potential redundancy, we performed a preliminary correlational analysis of all MRI metrics with WHR then focused our subsequent analyses on ICVF and ISOVF, the two complementary MRI metrics that were most strongly associated with WHR. Comparable results for the other 4 metrics are reported in the Supplemental Information Figure S2.”

12. What does it mean that "taste receptor activity is under-represented in the enrichment analysis for one but positively enriched in the analysis for the other MRI marker? If both are valid findings, please report both and not selectively report one.

We have clarified and balanced the reporting of the enrichment results for taste receptor activity. All the significant enrichment results are reported in full (not selectively) in Figure 2 and SI Figure S6.

13. Have inflammatory markers been investigated? For example, if IL-6 was associated with MRI markers in the same way as WHR, this would support the authors' interpretation of the data.

This interesting question has been raised as point R1, #5 above and we replicate our answer here. UK Biobank does not provide data on IL6 or other proinflammatory cytokines but it does provide CRP data as a broad index of systemic inflammation. As suggested we have now undertaken an additional analysis of associations between CRP and the microstructural MRI metrics to test the hypothesis that brain maps relating to CRP will be more similar to (more strongly correlated with) maps of obesity scaling with ISOVF (which we hypothesised represent potentially inflammatory effects of obesity on the brain) than to maps of obesity scaling with ICVF (which we hypothesised represent potentially causal effects of brain-mediated behaviour on obesity).

We have now addressed this point in the results and Discussion sections (as well as the abstract) and have added a new figure (Figure S13) which shows two main findings: (1) that there is a moderately strong association of CRP with microstructural brain metrics similar but weaker than WHR or BMI; and (2) that this relationship was statistically significantly stronger for the effect maps of CRP vs ISOVF compared to the maps of CRP vs ICVF as would be expected if the free water measured by ISOVF is related to inflammatory effects of obesity on the brain.

Reviewer #2 (Recommendations for the authors):Strengths1. The article takes an expansive, whole-brain approach to addressing the question of interest. While the methodology is simple (i.e., correlations), the rigor of the statistical method ensures that some of the results are plausible and are likely to be replicated. While I disagree with the range of techniques used, there is a comprehensive quality to them which for some reviewers and readers will be impressive.

We appreciate the reviewer’s positive comments about the rigour, robustness, comprehensiveness and plausibility of our work.

2. Finally, the figures are all high-quality and very well made, though incorrect or overreaching elements do cloud this otherwise rosy picture.

We thank the reviewer for appreciating the considerable effort that went into creating figures of a high standard and we hope that we have managed to address the issues raised.

Weaknesses3. In the introduction, the authors take a straightforward approach to address what is terrible literature, as I describe at length in a previous article. There are inconsistencies in how the brain is assessed, measures that are used to gauge obesity, the wide variance in sample size, poor statistical analysis of many brain regions, and so on. It is much less "pat" than the authors describe, and they really need to spend more time describing the non-significant and inconsistent findings over several brain regions described.

We agree that the literature is not entirely consistent and there are a number of methodological reasons that likely contribute to discrepancies between primary studies. The reviewer has not cited their previous article so we cannot be sure we have completely represented their perspective in our revision. However we have edited the Introduction to echo and highlight these legitimate concerns:

“To date, cross-sectional and longitudinal studies investigating effects of obesity on the brain have focused almost exclusively on macroscopic aspects of brain structure such as total grey matter volume and cortical thickness. Results in this field were often contradictory: although studies tended to report lower gray matter volume in relation to obesity, some have also observed null or positive associations as described in a meta-analysis by García-García et al. (2019), who noted that the likely reasons for this were heterogeneities in brain and obesity metrics, a wide variation in sample size, and poor statistical methodology.”

4. The transition to discussing DTI is also incomplete.

We have rewritten the relevant section of the Introduction to provide a more complete motivation for our focus on DWI-derived metrics:

“Unlike conventional diffusion MRI which models data acquired at a single diffusion weighting (shell), NODDI requires data collected at multiple different diffusion weightings (shells) and then exploits the diffusion characteristics that can be observed in different tissue compartments to quantify their respective volume fractions. In this model, diffusion is modelled as isotropic in free water, restricted within neurites, and hindered in the extracellular space, resulting in three microstructural metrics: intracellular volume fraction (ICVF) which captures the volume fraction occupied by neurites (axons and dendrites) but not cell bodies; orientation dispersion index (OD) which captures the spatial distribution of these processes; and isotropic volume fraction (ISOVF) which provides a measure of free water.”

5. Finally, I think it is odd to examine 180 cortical regions in a systematic fashion, rather than limiting analyses to larger regions and focusing on smaller regions based on initial analyses. This approach is faulty and prone to error, one which is exacerbated by FDR which looks great on paper but in practice is highly prone to type 1 error given enough comparisons.

We thank the reviewer for appreciating the considerable effort that went into creating figures of a high standard and we hope that we have managed to address the issues raised.

6. For methods, describe the final sample size in the main article. Do not relegate it to supplemental, as the sample size is really around 31,500 instead of 34,000 for most indices.

We agree with the reviewer that it is important to clarify the size of the analyzable sample and we have now done so in Abstract and Results.

7. I do not understand why people with high CRP values were excluded, yet scans were kept with participants that had all-cause dementia, heart failure, stroke, and a variety of other conditions that reliably show differences in the frontal lobe and other volumes.

Participants with high CRP (>10mg/L) were excluded due to potential effects on brain microstructure (see https://doi.org/10.1038/s41380-021-01272-1). The number of subjects with episodes of stroke or dementia in the UK Biobank data is less than 1%. However, your comment is well taken and in order to address these concerns we re-analysed the data excluding these subjects. We found a correlation of >0.99 between the resulting maps and the ones in the original analysis. We have now clarified this in the main text:

“In order to avoid spurious effects from pathologies causing systemic inflammation we also excluded subjects with high CRP (>10 mg/L). We repeated the analysis without subjects who had reported an episode of stroke or diagnosis of dementia, producing identical results.”

8. Only one scanner is described for UK Biobank but this is not correct. Three scanners were used, as described later on in the analysis pipeline. While the site can be added as a covariate, some studies looking at the effect of scanner units have been done and should be included.

Thank you for raising this point. We have now discussed the fact that there were three different sites using separate scanners, but of the same make and model and we reference some literature about the between site comparability in UK Biobank.

9. The use of the Allen Brain Atlas for gene expression data is unwise. Last I checked, a literal few participants in the same age range as the UK Biobank participants had been scanned using this technique.

It is true that the age range of the AHBA donors (24-57 years) is only partially overlapping with the participants in the UK Biobank (44-80 years). While future studies may be able to provide a comprehensive picture of whole brain gene expression as a function of age, for the time being we will have to accept the steady-state approximation that is the AHBA. Since this is an important limitation, we are now discussing it in text:

“It should also be mentioned that the age range of the AHBA donors (24-57 years) is only partially overlapping with the participants in the UK Biobank (44-80 years). Future studies will hopefully provide a more comprehensive picture of whole brain gene expression as a function of age so that the powerful strategy for linking transcriptional and imaging data that the AHBA dataset has enabled can be extended to gene expression datasets more closely aligned demographically with the neuroimaging dataset of interest. These and other methodological issues relating to alignment of AHBA gene expression data with MRI phenotypes have been rigorously reviewed in detail (Fornito et al. 2019, Arnatkeviciute et al. 2023).”

10. For results, BMI is not a good representation of central obesity and should be ignored. That it is relegated to supplemental is wise. WHR is better, but a correlation of r=0.428 is not "strong." It is a small-moderate effect size and should be treated appropriately.

We thank the reviewer for supporting our focus on WHR and we want to reiterate that we processed the data in parallel using both WHR and BMI and the interested reader can compare the results presented in the Supporting Information section.

This point is similar to point R1, #7 raised by Reviewer 1 above, where we have also quoted the additions to the main text justifying our preference for WHR as the principal marker of (visceral) obesity and discussing the metabolic importance of visceral fat:

“Given previous findings of significant association between macroscopic differences in brain structure and visceral obesity, we elected to report associations with WHR in the main text and report complementary results for BMI as a measure of whole body obesity in the SI.”

11. I find the use of volumetrics, DTI, and DTI-NODDI to be strange. For each region, technically, there should be 4 standard DTI metrics and 3 NODDI metrics.

We apologise for not being clearer in our rationale for focussing on neurite density and free water. We note that Reviewer 1 made similar comments (R1, #3 and R1, #8) previously and our response to this comment is substantially reproduced here. As we have now further clarified in the text, our goal was to use NODDI modelling of diffusion MRI to investigate associations of obesity with brain microstructure. We focussed on NODDI as this provides three (reasonably well validated and interpretable) indices of tissue microstructure. Of note, standard diffusion metrics such as mean diffusivity and fractional anisotropy grey matter have poor interpretability with respect to cortical microstructure and were only included to illustrate their relationship to the NODDI metrics.

We have rewritten the text to better motivate our choice of microstructural metrics and why these offer a new and complementary approach as copied below:

“However, changes in grey matter volume and cortical thickness can be driven by multiple different underlying processes and our understanding of the microstructural features that underpin this relationship remain largely unknown (Westwater et al., 2022). For example, it is currently not known whether obesity-associated differences in grey matter volume relate to changes in the size, shape or number of neurons e.g. neurite density or orientation dispersion within that region or alternately to differences in tissue water content.

[…]

Unlike conventional diffusion MRI which models data acquired at a single diffusion weighting (shell), NODDI requires data collected at multiple different diffusion weightings (shells) then exploits the diffusion characteristics that can be observed in different tissue compartments to quantify their respective volume fractions. In this model, diffusion is modelled as isotropic in free water, restricted within neurites, and hindered in the extracellular space resulting in three microstructural metrics: Intracellular Volume Fraction (ICVF) which captures the volume fraction occupied by neurites (axons and dendrites) but not cell bodies, Orientation Dispersion Index (OD) which captures the spatial distribution of these processes and isotropic volume fraction (ISOVF) which provides a measure of free water index).”

Our decision to restrict ourselves to two of the three NODDI metrics (i.e. excluding OD) was largely driven by a desire for brevity. The ICVF and OD maps were strongly correlated, so we elected to illustrate results for the more independent and complementary measures of ISOVF and ICVF in the main text and report the results on OD in SI rather than reporting the OD results in parallel to the closely related ICVF results in the main paper.

In response to the reviewer’s comments we have now repeated the analysis using PCA components rather than raw NODDI measures. This confirmed that ISOVF was an independent measure and that ICVF and OD were co-linear. We have now clarified this in the paper as copied below. However, we stuck with our decision to report NODDI metrics (rather than PCA components) as this is generally more easily interpretable for the general reader. The respective section in the main text is quoted below:

“As illustrated in Figure 1a, some of these metrics were strongly correlated, indicating that they represented similar aspects of the underlying cortical micro-structure or tissue composition. For example, FA, OD and ICVF metrics of neurite density were more strongly correlated with each other than with ISOVF, which is typically interpreted as a marker of tissue free water rather than cytoarchitectonics (Kamiya et al., 2020).

To address this potential redundancy, we performed a preliminary correlational analysis of all MRI metrics with WHR, then focused our subsequent analyses on ICVF and ISOVF, the two complementary MRI metrics that were most strongly associated with WHR. Comparable results for the other 4 metrics are reported in the Supplemental Information Figure S2.”

12. Figure 1 is slick. I do not see the point of having a correlation matrix like Figure 1a.

We thank the reviewer for appreciating the considerable effort that went into creating figures of a high standard. Figure 1a is meant to illustrate the correlated nature of the available imaging metrics. Specifically it demonstrates that the free-water metric ISOVF is essentially orthogonal to the measures of neurite density ICVF and orientation OD, whereas the latter two are forming a cluster together with fractional anisotropy FA. We have now added this explanation to the legend of Figure1:

“a) Correlation matrix for six macro- and micro-structural MRI metrics demonstrating that ISOVF (free-water) is essentially orthogonal to ICVF (neurite density) and OD, which instead form a cluster with FA.”

13. For Figures 1b and 1c, as well as Figures 1d and 1e, what I am struck by is the small correlation found in the areas that show the strongest association. That higher WHR is related to greater ICVF in the hippocampus is important, for example, but a correlation of 0.32 is hardly something to worry about.

The reviewer’s opinion that a correlation of 0.32 is “hardly something to worry about” is entirely subjective. A correlation of 0.32 means that about 10% of hippocampal variance is related to WHR, which is not trivial, and correlations greater than 0.3 are generally regarded as moderate (not small) in the statistical literature, e.g., https://doi.org/10.1016/j.paid.2016.06.069.

14. I demand a range of correlation matrices besides what looks best. Further, I find the associations with the occipital lobe hard to believe. This region rarely shows significant associations with obesity. Likewise, I find the frontal and in particular prefrontal areas showing no association to be surprising. In comparing all participants aged 40-70 years, perhaps comparisons by binned age groups would be better.

It is not clear exactly what the reviewer is “demanding” here. As far as we know, all the relevant correlation matrices are reported in full in the main text or supplementary information; we would be happy to include additional results if the reviewer clearly specified what they felt was missing.

The reviewer’s insinuations, here and elsewhere in their remarks, that we have selectively reported “what looks best” is not well documented by the granular detail of their comments and is, frankly, unjustified and disrespectful. We have reported the results of an unusually comprehensive analysis, including multiple sensitivity analyses and statistical controls to ensure the robustness of our findings. The reviewer may find it “hard to believe” the presence or absence of certain results they expected in the occipital or frontal lobes but, as stated, this is no more than their personal opinion. We would be happy to include further discussion of expected and unexpected results if the reviewer could kindly point us to the prior evidence on which their expectations are based. It is not clear to us why “comparisons by binned age groups would be better” – it would certainly dramatically increase the number of statistical tests and increase the risk of type 1 error, so we have therefore not elected to adopt this suggestion.

15. I do not see the point of using the Allen Brain Atlas, as described. It is not only non-representative of gene expression given its very small sample size, but a lack of suitability to the British population versus the handful of younger to a literal few older Americans used to make the maps. Use Bonferroni correction to truly take into account the number of comparisons made and get back to me.

As we pointed out in the original paper, and have amplified in this revision, the Allen Brain Atlas is not perfect but it is the only available dataset on human brain gene expression that is available and appropriate for analysis of whole genome transcriptional profiles spatially co-located with MRI phenotypes in the UKB cohort. The high value attached to the Allen Brain Atlas by the neuroscience research community is attested by the fact that it has been cited between 1400-2400 times (depending on metric), and many of the neuroimaging papers using the Allen Brain Atlas have themselves been highly cited. The limitations of the existing dataset, and the methodological issues involved in aligning it with MRI phenotypes, have been well-recognised and discussed in the field (as noted in our response to R2 #9).

This comment: “use Bonferroni correction to truly take into account the number of comparisons made and get back to me” is again phrased in an intemperate and disrespectful tone. We note that all the analyses have been corrected for multiple comparisons using the false discovery rate which is a well-recognised and very widely used method for this purpose. Without more granular and well-founded arguments in favour of using the alternative Bonferroni procedure (which is well acknowledged to inflate type 2 error rates: https://doi.org/10.1136/bmj.316.7139.1236), we have elected not to adopt this suggestion.

16. What I most strongly object to is the lack of actual metrics (i.e., r, R^2, Cohen's D, etc.) describing any of the correlations. We are shown plots but are not given correlations in many cases (e.g., Figure 4). This makes for more slick figures, but to what end and for what purpose? I am not going to win over by glam and type-setting.

We have added relevant or clarified quantitative data as appropriate throughout the text, figures, figure legends and supplemental information. The figures were not intended to be “slick” or “glam” (again, somewhat disrespectful language for a scientific review); they were intended to be high quality and engaging representations of an extensive analysis of a large and complex dataset. We are always happy to take on board constructive and specific feedback about how the quality of the figures in this paper could be further improved.

17. The discussion is off-putting. To be clear, the authors focused on two DTI-NODDI indices and relegated three other metrics to supplemental because they were less impressive. This is not a hypothesis. There was no a priori determination that ISOVF and ICVF would show impressive results.

The rationale for focusing on two NODDI metrics was set out in the Introduction and has been further clarified in this revision (see our response to the very similar point made previously by this reviewer, #11). The reviewer’s insinuation that we “relegated three other metrics to supplemental because they were less impressive” is, again, unjustified and disrespectfully phrased.

18. This was the result of an FDR-corrected smorgasbord of analyses that the authors hoped would win over reviewers because of how nice everything looked. I do machine learning and limited computational biology techniques every day. I am not impressed. The rest of the discussion is based on a corpus of findings that are either not appropriate (e.g., Allen Brain Atlas) or many series of FDR-corrected findings that I do not believe.

As detailed above, we have responded substantively to the reviewer’s concerns about the use of Allen Brain Atlas data and FDR correction to control Type-I error rate, both of which are widely used methods in the neuroimaging literature. As to the reviewers opinion that we used “a smorgasbord of analyses that [we] hoped would win over reviewers because of how nice everything looked”, we would merely highlight again the disrespectful nature of this comment.